# Fault-Source-Based Probabilistic Seismic Hazard and Risk Analysis for Victoria, British Columbia, Canada: A Case of the Leech River Valley Fault and Devil's Mountain Fault System

**Katsuichiro Goda \*** and **Andrei Sharipov**

Department of Earth Sciences, Western University, London, ON N6A 5B7, Canada; asharip@uwo.ca
\* Correspondence: kgoda2@uwo.ca; Tel.: +1-519-661-2111 (ext. 83189)

**Abstract:** This study develops a fault-source-based seismic hazard model for the Leech River Valley Fault (LRVF) and the Devil's Mountain Fault (DMF) in southern Vancouver Island, British Columbia, Canada. These faults pose significant risks to the provincial capital, Victoria, due to their proximity and potentially large earthquake magnitudes. To evaluate the effects of including these faults in probabilistic seismic hazard analysis and city-wide seismic loss estimation for Victoria, a comprehensive sensitivity analysis is conducted by considering different fault rupture patterns and different earthquake magnitude models, as well as variations in their parameters. The aim is to assess the relative contributions of the LRVF-DMF system to the overall seismic hazard and risk in Victoria at different return periods. The consideration of the LRVF-DMF system as a potential seismic source increases the seismic risk assessment results by 10 to 30%, especially at the high return period levels. The sensitivity analysis results highlight the importance of determining the slip rate for the fault deformation zone and of specifying the earthquake magnitude models (e.g., characteristic versus truncated exponential models). From urban seismic risk management perspectives, these nearby faults should be considered critical earthquake scenarios.

**Keywords:** probabilistic seismic hazard analysis; portfolio seismic loss estimation; Leech River Valley Fault; Devil's Mountain Fault; sensitivity analysis; wooden houses; critical scenarios; seismic risk management

## 1. Introduction

Probabilistic seismic hazard and risk assessments and their uncertainty quantification are essential for making informed decisions regarding seismic risk mitigation actions and for enhancing disaster preparedness [1,2]. An earthquake catastrophe model offers an effective computational platform for calculating the economic loss due to earthquake disasters and has become an indispensable tool for the insurance and reinsurance industry [3]. The state-of-the-art earthquake catastrophe models can produce exceedance probability curves and critical loss scenarios for building portfolios [4]. Effective seismic risk management, informed by sound disaster risk reduction strategies, will safeguard people and their assets and promote the sustainable development of the built environment.

Southwestern British Columbia is within an active seismic region of the Cascadia subduction zone [5]. Vancouver and Victoria, which are the economic and political centers of the province, are exposed to significant seismic risks [6–8], originating from three major sources: shallow crustal earthquakes, deep inslab earthquakes, and megathrust Cascadia subduction earthquakes. Recently, Goda et al. [9] conducted a city-wide seismic risk assessment of single-family wooden houses in Victoria by considering a comprehensive building-by-building exposure model, the national seismic hazard model developed by the Geological Survey of Canada (GSC) [10], and seismic fragility functions, based on rigorous nonlinear dynamic analysis of structures and ground-motion record selection [11]. A full consideration of stochastic event scenarios in probabilistic seismic risk analysis

facilitated the identification of critical loss scenarios from regional seismic risk perspectives and generated a set of integrated seismic hazard and risk maps that correspond to the identified scenarios.

New geological evidence and geophysical investigations of active faults near Victoria reveal the Leech River Valley Fault (LRVF) and the Devil's Mountain Fault (DMF) as potential seismic hazard sources. The steeply dipping LRVF has been identified as a potential active fault in southern Vancouver Island based on paleoseismic surface-rupture traces in the Holocene [12] and microseismic activity [13]. The southeastern tip of the LRVF is within a few kilometers from Victoria. On the other hand, Barrie and Greene [14] mapped the DMF zone spanning from Washington State to Vancouver Island based on high-resolution submarine geophysical surveys. Their investigations indicated that the fault zone was active in the Holocene and has the potential to produce a strong earthquake. They also suggested that the DMF and the LRVF may be part of the same fault system. From seismic hazard assessment perspectives, it is possible that the two faults rupture synchronously, resulting in a larger earthquake magnitude [15], and such an event could pose immense threat to people and assets in Victoria due to its proximity.

Two recent studies investigated the seismic hazard implications of the LRVF-DMF system for Victoria [15,16]. Halchuk et al. [15] adopted the characteristic magnitude model [17] for the LRVF-DMF system and performed a fault-source-based probabilistic seismic hazard analysis (PSHA) by integrating the LRVF-DMF model into the preliminary version of the GSC2020 seismic hazard model. They concluded that the increase due to the LRVF-DMF fault source is relatively small (5%) for peak ground acceleration (PGA) at the annual probability of exceedance of $4 \times 10^{-4}$ or the return period of 2475 years. In contrast, Kukovica et al. [16] derived magnitude–recurrence relationships that are applicable to the LRVF zone based on available local seismic catalogs and assessed the seismic hazard for Victoria due to the LRVF via an area-source-based PSHA, ignoring ruptures originating from the DMF. They concluded that the inclusion of the LRVF in the GSC2015 seismic hazard model results in a 9% increase in PGA at the return period of 2475 years. The above-mentioned two PSHA studies are deemed to be limited because they did not consider the full extent of uncertainties associated with different fault rupture models, as well as the model parameters. More importantly, these studies mainly focused upon the seismic hazard contributions of the LRVF-DMF source at the 2475-year return period level and did not address more extreme seismic excitations, nor potential seismic risks due to this nearby fault source. However, considering longer return period levels and assessing seismic risk to the existing building stock in Victoria are of primary importance from seismic risk management perspectives and, therefore, need to be carried out.

The main objectives of this work are two fold. First, we develop a comprehensive fault-source-based probabilistic seismic hazard model for the LRVF-DMF system by considering different spatial rupture patterns and different types of the earthquake magnitude model, as well as variations of their parameters. An extensive sensitivity analysis is performed to critically assess the relative contributions of the LRVF-DMF system to the overall seismic hazard in Victoria at different return period levels, exceeding the 2475 years. The PSHA results for the LRVF-DMF system are compared with those based on the GSC2015 model. Since the rupture of a large earthquake from the LRVF-DMF system is rare (in the order of several thousands of years), the potential hazard and risk impact from the LRVF-DMF system cannot be fully quantified by looking at the seismic hazard level corresponding to the 2475-year return period alone. In other words, relatively small contributions of the LRVF-DMF system at this return period do not necessarily mean that the hazards and risks due to this fault system are low and, thus, are negligible. Second, we evaluate the city-wide seismic risk to residential buildings in Victoria. We compare the relative contributions of the LRVF-DMF system in terms of city-wide seismic risk metrics with those from other hazard sources (e.g., crustal, interface, and inslab earthquakes). Based on the sensitivity analysis results, we aim to provide important insights for seismic risk management in Victoria. In short, the significance of this work is that comprehensive

sensitivity analyses of the fault-source-based PSHA and city-wide seismic loss estimation for Victoria, due to the potentially devastating LRVF-DMF system, are carried out to identify the most influential fault parameters and to evaluate the effects of the potentially devastating LRVF-DMF system on urban seismic risk management. In Section 2, the study area is introduced. Section 3 presents the methodology of the current study, including the fault-source-based PSHA, and the exposure-vulnerability models for residential buildings in Victoria. In Section 4, a series of sensitivity analyses of the LRVF-DMF system are carried out by considering different variations of the seismic hazard model components, whereas in Section 5, quantitative seismic risk assessments are performed, accompanied by the sensitivity analyses related to the characterization of the LRVF-DMF system. Finally, a set of conclusions is drawn from the viewpoints of advancing disaster preparedness and sustainable development.

## 2. Study Area

### 2.1. Leech River Valley Fault and Devil's Mountain Fault

Seismicity in southwestern British Columbia is complex. In the Cascadia subduction zone, the Juan de Fuca Plate subducts underneath the North American Plate with their relative plate motions of 40 mm/year, as measured by GPS velocities [18]. The locked shallow portion of the plate interface results in a megathrust subduction event of moment magnitude ($M_w$) 8 and greater (e.g., 1700 $M_w$9 earthquake), while the deeper portion of the subducting oceanic plate generates damaging inslab events beneath Puget Sound (e.g., 1949, 1965, and 2001 earthquakes) [5]. These seismic sources are recognized in the GSC seismic source zone model, as shown in Figure 1a. For instance, the CIS (Cascadia Interface Source) zone corresponds to $M_w$8 to $M_w$9 megathrust Cascadia subduction events, whereas the GTP (Georgia Strait/Puget Sound) zone is defined to characterize deep inslab events. In the continental plate, diffused crustal seismicity is present. Figure 1b shows the local seismicity in southern Vancouver Island [10]. In the GSC hazard model, the PGT (Puget Sound shallow) source mainly captures shallow crustal seismicity in the vicinity of Victoria (Figure 1a).

Paleoseismic and geomorphological studies identified several active faults in this region [19]. Among those, concerning seismic hazard and risk in Victoria, the LRVF-DMF system is the most critical one due to its proximity to Victoria [12–14] (Figure 1b,c). The LRVF is the north-dipping reverse fault zone with a dip angle of 60° to 70° and total length of 60 to 70 km [15,16], and hosted three surface-rupturing earthquakes over the last 9000 years [12]. Morell et al. [12] suggest that the Holocene slip rate of at least 0.2–0.3 mm/year can be considered for the LRVF. On the other hand, the DMF is a north-dipping deformation zone that extends from Washington State to south of Victoria in the Strait of Juan de Fuca. There are at least two pieces of paleoseismic evidence of moderate-to-large earthquakes in the Holocene [14]. The estimated vertical slip rates of the DMF range between 0.05 and 0.31 mm/year with the representative estimate of 0.18 mm/year [19]. Moreover, the LRVF and the DMF can be viewed as a multi-segment fault system [14], with the junction point very near Victoria (Figure 1c), which can rupture synchronously [15].

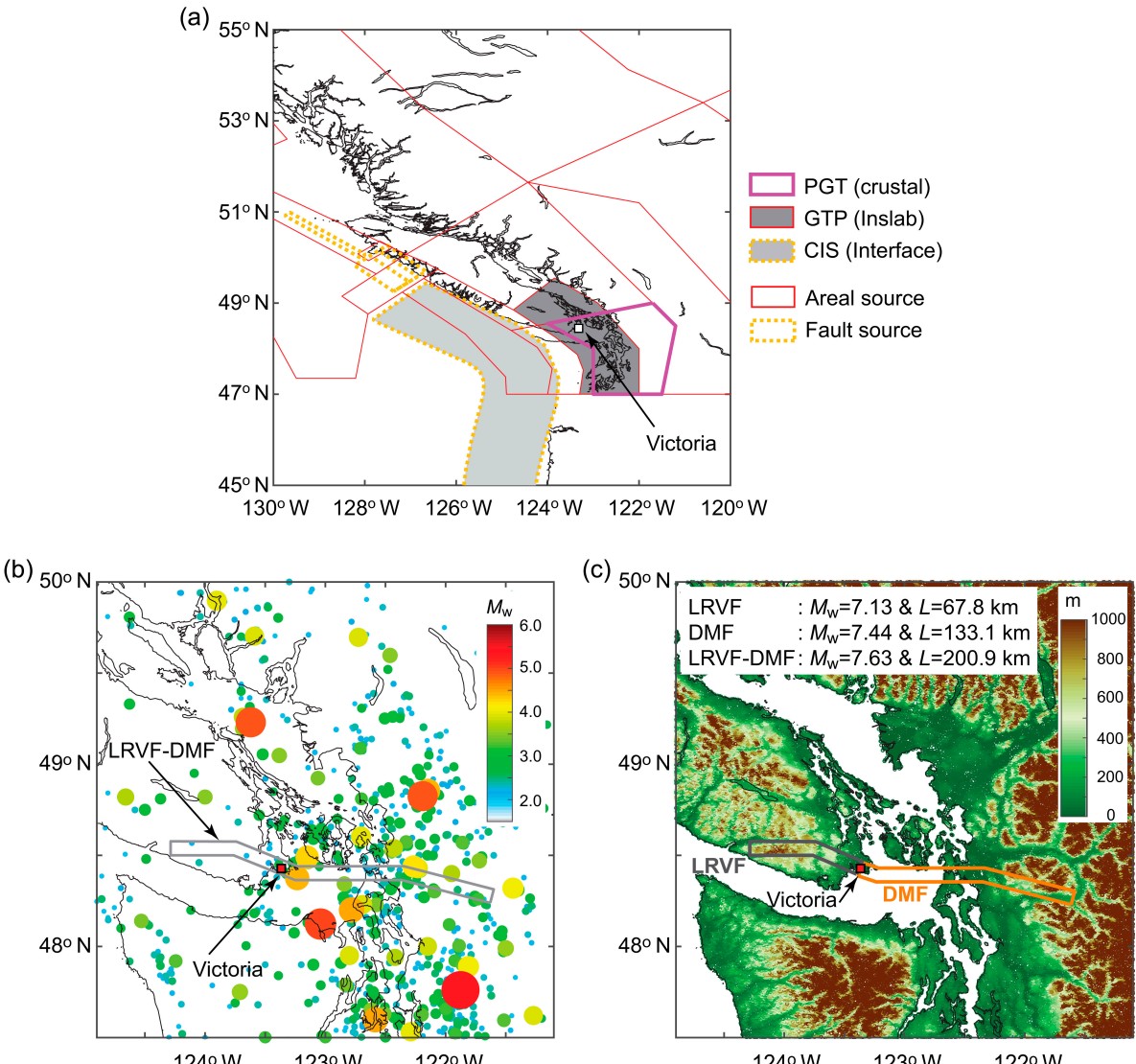

**Figure 1.** (**a**) Seismic source zone model by the Geological Survey of Canada, (**b**) local seismicity near Victoria, and (**c**) fault geometry of the LRVF-DMF system. In (**c**), the SRTM digital elevation model is shown in the background.

### 2.2. Residential Wooden Houses in Victoria

Victoria has variable subsurface geology, mainly consisting of Holocene organic soils, glaciomarine clays, glacial till, and bedrock. The relative site amplification maps produced by [20] indicate that the site conditions for the City of Victoria can be broadly described as stiff-to-soft soils (site class C to D), while several small pockets of areas are identified as very soft soils (site class E to F). In terms of the near-surface site parameter $V_{S30}$ (i.e., average shear-wave velocity in the upper 30 m), the former corresponds to $V_{S30}$ of 180 to 760 m/s, whereas the latter corresponds to $V_{S30}$ less than 180 m/s.

Victoria is the provincial capital of British Columbia and is the seventh most densely populated area in Canada. There are many masonry buildings and historical buildings (mainly commercial and governmental occupancy) in the downtown core of Victoria, whereas the majority of residential buildings are wooden buildings, and some of them are old, dating back to approximately 1900. The building exposure database for wooden houses in Victoria is available from the BC Assessment database (https://www.bcassessment.ca/). The original database, created as of June 2013, includes 13,933 buildings with various structural typologies, such as wood frames for single-family and multi-family residential use, steel frames, reinforced concrete frames, masonry-bearing walls, and wood-steel

hybrid structures. Among these, 6683 wooden houses for single-family residential use with a total floor area of less than 5000 ft$^2$ are selected for city-wide seismic risk assessments in Victoria (see Section 3.2 for more details). In terms of building assessment values, the selected buildings account for 22.7% and 36.6% of the entire and residential building stock in Victoria, respectively.

## 3. Methodology

A standard earthquake catastrophe modelling approach is adopted to evaluate seismic hazard and risk in Victoria. The main model elements are: (i) seismic hazard model, (ii) building exposure model, and (iii) seismic vulnerability model (Figure 2). These can be integrated to assess seismic hazard and risk quantitatively and to perform sensitivity analyses. For the seismic hazard modelling, we focus on the fault rupture modelling of the LRVF-DMF system based on the stochastic source approach [21] and the earthquake magnitude modelling based on the seismic moment rate balancing approach [17]. In this paper, we do not provide details of the seismic hazard model and analysis for other seismic sources in southwestern British Columbia. The seismic hazard model for other sources is based on the GSC2015 model [10]. This model has been independently implemented using a simulation-based PSHA (through the generation of stochastic event sets), and its accuracy with respect to the GSC2015 model has been verified to be within 2–3% [11]. Therefore, the developed seismic hazard model for the LRVF-DMF system in Section 3.1 can be seamlessly incorporated into the full GSC2015 seismic hazard model. Moreover, the descriptions of the building exposure model and seismic vulnerability model for wood-frame houses in Victoria in Sections 3.2 and 3.3 are kept concise, because the full details are available in [9,11].

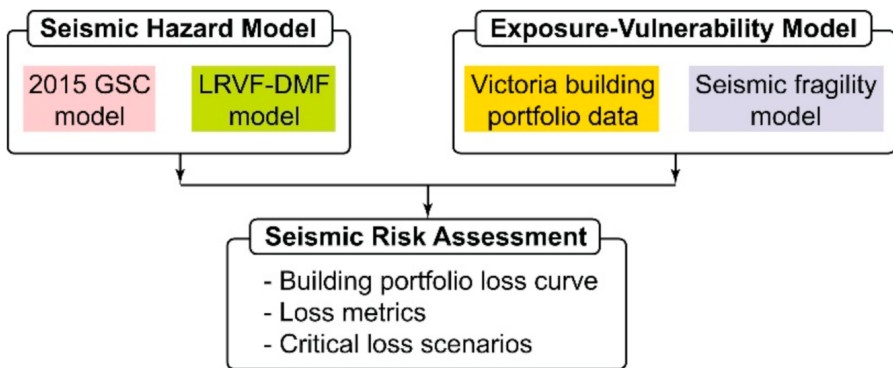

**Figure 2.** An overview of quantitative seismic hazard and risk assessments.

### 3.1. Seismic Hazard Model for the Leech River Valley Fault and Devil's Mountain Fault System
#### 3.1.1. Fault Rupture Model

To capture different spatial rupture patterns, two fault models, i.e., synchronous and segmented rupture cases, are considered for the LRVF-DMF system. The fault geometry of the LRVF-DMF system that was considered by [15] is adopted, and this fault geometry for the synchronous and segmented rupture cases is shown in Figure 1b,c, respectively. When the synchronous rupture case is considered, ruptures can occur within the combined fault zone shown in Figure 1b. On the other hand, for the segmented case, earthquake ruptures can occur independently within individual fault zones of the LRVF and the DMF and their spatial rupture extents do not go beyond the fault junction boundary (Figure 1c). We recognize that the fault rupture patterns involve significant uncertainty. With the currently available paleoseismic and geomorphological evidence and data, it is not possible to determine the likelihoods of these fault rupture patterns. For instance, Halchuk et al. [15] considered that the relative likelihood of the synchronous versus the segmented rupture scenarios is 10% versus 90%. In our study, we treat this as epistemic uncertainty and model it explicitly as part of the logic tree for the LRVF-DMF system (see Section 3.1.4).

The fault lengths of the LRVF, the DMF, and the LRVF-DMF system are 67.8, 133.1, and 200.9 km, respectively. Using the earthquake source scaling relationship [22], these fault lengths correspond to median estimates of $M_w 7.13$, $M_w 7.44$, and $M_w 7.63$, respectively. The seismogenic width of the fault zone (applicable to all three cases) is determined based on the dip angle of 70° [15] and is set to 25 km in this study. This fault width is greater than the width assumed by [15] ($\approx$16 km) but is consistent with other studies [13,16,22]. The wider fault dimension is to accommodate uncertain fault rupture geometry and position within the potential deformation zone via a stochastic source modelling approach [21].

Earthquake ruptures originating from the LRVF-DMF system are represented by rectangular finite-fault sources on the fault rupture plane (Figure 1b,c). Specifically, for a given moment magnitude of an earthquake rupture, the fault length and width are generated from the statistical earthquake source scaling relationships [22] and the fault geometry is floated within the fault zone boundary. The stochastic source approach [21] allows for the consideration of spatial uncertainty associated with the rupture size and location as a function of earthquake magnitude, which is particularly important for the case of the LRVF-DMF system, due to its proximity to Victoria. These stochastic finite-fault sources are used to calculate the source-to-site distances and simulate ground-motion intensity fields in seismic hazard and risk assessments (see Section 3.1.3).

### 3.1.2. Earthquake Magnitude Model

The occurrence rates and earthquake magnitudes are characterized by considering a combination of the truncated exponential model (or more conventionally known as the Gutenberg–Richter relationship) and the characteristic model. The starting point in characterizing the magnitude–recurrence relationship for both magnitude models is to specify the slip rate within the fault zone. The slip rate serves to control how active the fault system is in terms of seismic moment release rate:

$$\dot{M}_0 = \mu A_f S_r \tag{1}$$

where $\mu$ is the shear modulus, $A_f$ is the area of the fault zone, and $S_r$ is the slip rate of the fault zone. On the other hand, different earthquake magnitude models characterize how the seismic moment release is distributed over the earthquake magnitude range.

The model formulations introduced by [17] facilitate the consistent seismic moment release from the two alternative magnitude models. The probability density function for the characteristic magnitude model, which consists of the exponential part and the characteristic part, is expressed as [23]:

$$f(m) = \begin{cases} 0 & \text{for } m < M_{min} \\ \frac{\beta \exp(-\beta(m - M_{min}))}{(1+C)[1 - \exp(-\beta(M_{max} - M_{min} - \Delta m_2))]} & \text{for } M_{min} \leq m < M_{max} - \Delta m_2 \\ \frac{\beta \exp(-\beta(M_{max} - M_{min} - \Delta m_1 - \Delta m_2))}{(1+C)[1 - \exp(-\beta(M_{max} - M_{min} - \Delta m_2))]} & \text{for } M_{max} - \Delta m_2 \leq m \leq M_{max} \\ 0 & \text{for } m > M_{max} \end{cases} \tag{2}$$

where the constant $C$ represents the total probability mass for the characteristic part and is given by:

$$C = \frac{\beta \exp(-\beta(M_{max} - M_{min} - \Delta m_1 - \Delta m_2))}{1 - \exp(-\beta(M_{max} - M_{min} - \Delta m_2))} \Delta m_2 \tag{3}$$

In the above equations, $\beta = b \times \ln(10)$, where $b$ is the slope parameter of the Gutenberg–Richter relationship; $M_{min}$ and $M_{max}$ are the minimum magnitude and maximum magnitude for the fault source, respectively; $\Delta m_1$ is the magnitude interval that is used to specify the probability density value for the characteristic part; $\Delta m_2$ is the magnitude interval for the characteristic part. The interpretation of Equation (2) is that, in the magnitude range between $M_{min}$ and $M_{max} - \Delta m_2$, the magnitude distribution follows the exponential distribution, whereas in the magnitude range between $M_{max} - \Delta m_2$ and $M_{max}$, the magnitude distribution follows the uniform distribution (i.e., constant probability density). It is noted

that Equation (2) accommodates the truncated exponential magnitude model by setting $\Delta m_2 = 0$ (i.e., $C = 0$). The seismic activity rate for the exponential part of $f(m)$ is given by [23]:

$$\alpha_{exp} = \frac{\mu A_f S_r [1 - \exp(-\beta(M_{max} - M_{min} - \Delta m_2))]}{K M_0^{max} \exp(-\beta(M_{max} - M_{min} - \Delta m_2))} \tag{4}$$

while the seismic activity rate for the characteristic part of $f(m)$ is given by:

$$\alpha_{char} = \alpha_{exp} \frac{\beta \Delta m_2 \exp(-\beta(M_{max} - M_{min} - \Delta m_1 - \Delta m_2))}{1 - \exp(-\beta(M_{max} - M_{min} - \Delta m_2))} \tag{5}$$

In Equations (4) and (5), $M_0^{max}$ is the seismic moment that corresponds to the maximum magnitude $M_{max}$, and the constant $K$ is given by:

$$K = \frac{b 10^{-1.5\Delta m_2}}{1.5 - b} + \frac{b \exp(\beta \Delta m_1)(1 - 10^{-1.5\Delta m_2})}{1.5} \tag{6}$$

The characteristic magnitude model requires the following model parameters: $\mu$, $A_f$, $S_r$, $b$ (or $\beta$), $M_{min}$, $\Delta m_1$, $\Delta m_2$, and $M_{max}$, noting that these parameters are sufficient for the truncated exponential model as it is a special case of the characteristic model. In this study, the shear modulus is set to 35 GPa (deterministic), whereas the fault zone areas for the three rupture cases are calculated from the fault length and width, shown in Figure 1b,c. The slip rate for the fault zone is highly uncertain. As a base case, the slip rate is represented by a discrete random variable, taking values of 0.25, 0.15, and 0.35 mm/year with weights of 0.68, 0.16, and 0.16, respectively (i.e., implemented as logic-tree branches, as shown in Figure 3a). These slip values are also considered by [15] and are consistent with the current best estimates of this uncertain quantity [12,19]. The $b$ value is adopted from the areal seismic source zone PGT and is represented by a discrete random variable, taking values of 0.796, 0.730, and 0.862 with weights of 0.68, 0.16, and 0.16, respectively (Figure 3a). The minimum magnitude for the LRVF-DMF system is set to $M_{min}$ = 6.0 (deterministic), which is consistent with [15], while the values of $\Delta m_1$ and $\Delta m_2$ are set to 1.0 and 0.5 (deterministic), respectively, which are the original values suggested by [17] and are also considered by [15]. The maximum magnitude is determined by first evaluating the magnitude value that corresponds to the fault length and width of the rupture zone using the scaling relationship by [22] (see Figure 1c for the fault dimensions and the calculated magnitude values) and by adding a half of the characteristic magnitude range (i.e., $\Delta m_2 / 2 = 0.25$). For instance, the $M_{max}$ value for the LRVF is calculated as $(\log_{10}(67.8 \times 25) + 3.486)/0.942 + 0.25 = 7.13 + 0.25 = 7.38$. In addition, considering the major epistemic uncertainty of $M_{max}$, the logic-tree branch is introduced by considering the $M_{max}$ shifts of 0.0, $-0.15$, and 0.15 with weights of 0.6, 0.3, and 0.1, respectively (Figure 3a). In short, for a given earthquake rupture scenario (i.e., either synchronous LRVF-DMF rupture or segmented LRVF and DMF ruptures), two earthquake magnitude models (i.e., characteristic and truncated exponential models) can be specified, and for each magnitude model, there are 27 variations due to discrete values assigned to slip rate, $b$ value, and $M_{max}$ shift (Figure 3a). It is noted that the relative likelihoods of the truncated exponential model versus the characteristic model (for a given rupture case) are difficult to determine explicitly and both models are considered to be applicable in conducting PSHA [24,25]; therefore, we treat this epistemic uncertainty as part of the logic tree for the LRVF-DMF system (see Section 3.1.4).

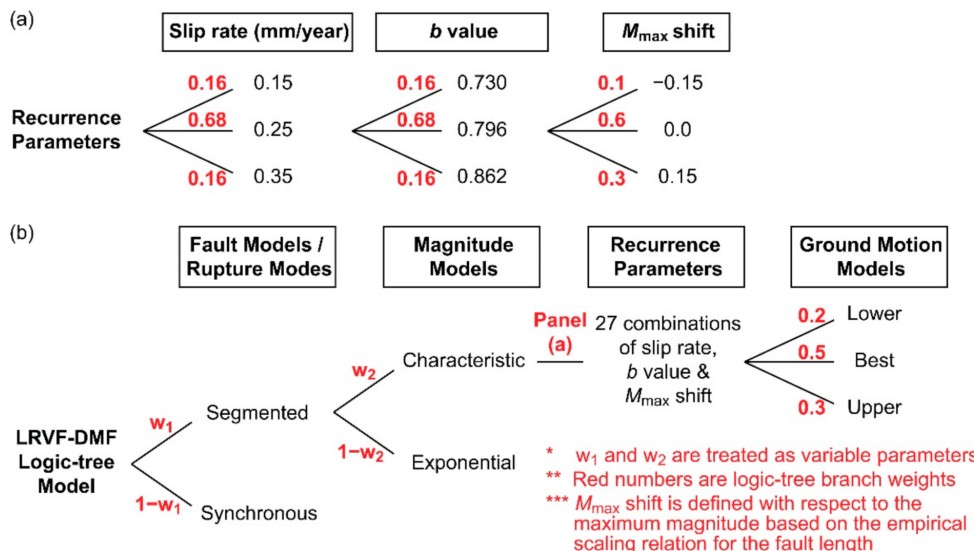

**Figure 3.** (**a**) Logic tree for the recurrence parameters and (**b**) logic tree for the LRVF-DMF system.

### 3.1.3. Ground-Motion Model

The fault models and the magnitude models developed in Sections 3.1.1 and 3.1.2 describe the fault source characteristics of the LRVF-DMF system and can be used to generate finite-fault stochastic event sets for PSHA calculations. Subsequently, ground-motion intensities, such as PGA and SA, need to be evaluated for the generated stochastic events. In this study, the same set of ground-motion models implemented in the GSC seismic hazard model [10] is considered for the LRVF-DMF system. The ground-motion models incorporate epistemic uncertainty by implementing three alternative models (i.e., best, lower, and upper) with respective weights of 0.5, 0.2, and 0.3 (Figure 3b). The three prediction models holistically represent different ground-motion models that are applicable to shallow crustal earthquakes in southwestern British Columbia [26]. It is noted that different sets of the ground-motion models are adopted for different earthquake types (e.g., deep inslab and megathrust interface events).

For the seismic hazard sensitivity analysis associated with the LRVF-DMF system (Section 4), a single site (latitude = 48.428° N and longitude =123.366° W) is considered. Its site condition is set to the reference ground condition for seismic hazard mapping in Canada, which corresponds to the near-surface site characterized by $V_{s30}$ = 450 m/s [27].

For the seismic risk sensitivity analysis (Section 5), 221 grid points that cover the City of Victoria at 0.005° intervals are set up, as shown in Figure 4. To assess the aggregate seismic risk for the building portfolio, spatially correlated ground-motion fields are generated for individual stochastic events by considering the median ground-motion models and the spatial correlation model [28]. This facilitates the consideration of realistic spatial variations of ground-motion intensity within the city. At the 221 grid locations, the site conditions represented by the $V_{s30}$ values obtained from [29] are accounted for by adjusting the local site amplification factors [27]. The majority of the locations are assigned with $V_{S30}$ between 200 and 450 m/s, falling into site classes C to D. This is consistent with Monahan et al.'s baseline map for Victoria [20], although it lacks very soft soil sites in several local areas.

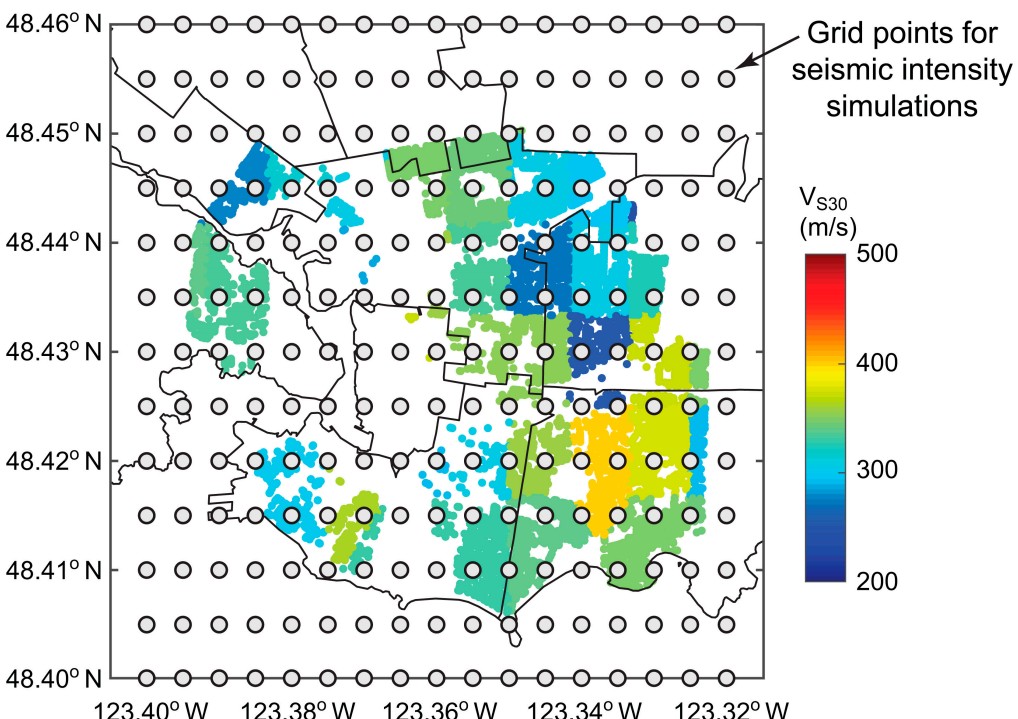

**Figure 4.** Average shear-wave velocity map at building locations based on the global $V_{S30}$ map. Individual wooden houses are represented by the dots.

3.1.4. Logic Tree for Uncertainty Characterization

The uncertainty characterization is the critical element in PSHA. With the focus on the LRVF-DMF system impacting the seismic hazard and risk in Victoria, a comprehensive logic tree is developed. By recognizing the major epistemic uncertainty related to the fault models (synchronous versus segmented rupture scenarios) and the earthquake magnitude models (characteristic versus exponential models), the weights of the logic-tree branches for these two aspects, $w_1$ and $w_2$, are regarded as assignable and their effects on PSHA results are investigated as part of sensitivity analysis by varying from 0.0 to 1.0 with a 0.2 increment. In contrast, the magnitude–recurrence relationship is varied through the 27 combinations of three fault-source variables, i.e., slip rate, $b$ value, and $M_{max}$ shift (Section 3.1.2), and the ground-motion model is varied based on the three alternative models (Section 3.1.3). The logic-tree structure for the LRVF-DMF system is shown in Figure 3b. For the base case (see Sections 4 and 5), the weights $w_1$ and $w_2$ are set to 0.5 (i.e., equal weighting for the synchronous and segmented rupture scenarios and for the two magnitude models).

*3.2. Exposure Model for Residential Wooden Buildings in Victoria*

For city-wide seismic loss estimation, 6683 single-family wooden houses, which are mainly located outside of the downtown core and bay area, are considered. This building inventory is identical to [9], thus the details are not repeated herein. Figure 5a shows the spatial distribution of the selected houses, whereas Figure 5b shows a histogram of the selected houses in terms of year of construction. For seismic vulnerability assessments, the selected wooden buildings are represented by the four UBC-SAWS models [30,31]. Differences in the seismic capacities of the four models, i.e., Houses 1 to 4, can be attributed to different structural configurations of shear walls (e.g., horizontal board, gypsum wallboard, plywood, oriented strand board, and exterior stucco), and their overall seismic capacities can be ordered as House 1 > House 2 ≈ House 3 > House 4 [31]. Based on the available information on the history of seismic design codes [7,32], a house model type is assigned to each of the selected buildings in Victoria based on the construction year: House 1 for

the years after 2005, House 2 for the years between 1995 and 2005, House 3 for the years between 1975 and 1995, and House 4 for the years before 1975. This classification scheme is indicated in Figure 5b. The majority of the residential houses in Victoria (=5976) are categorized as House 4 (i.e., limited seismic resistance), whereas the numbers of seismically resistant houses are relatively small with 119, 399, and 189 for House 1 (high seismic resistance), House 2 (moderate seismic resistance), and House 3 (moderate seismic resistance), respectively.

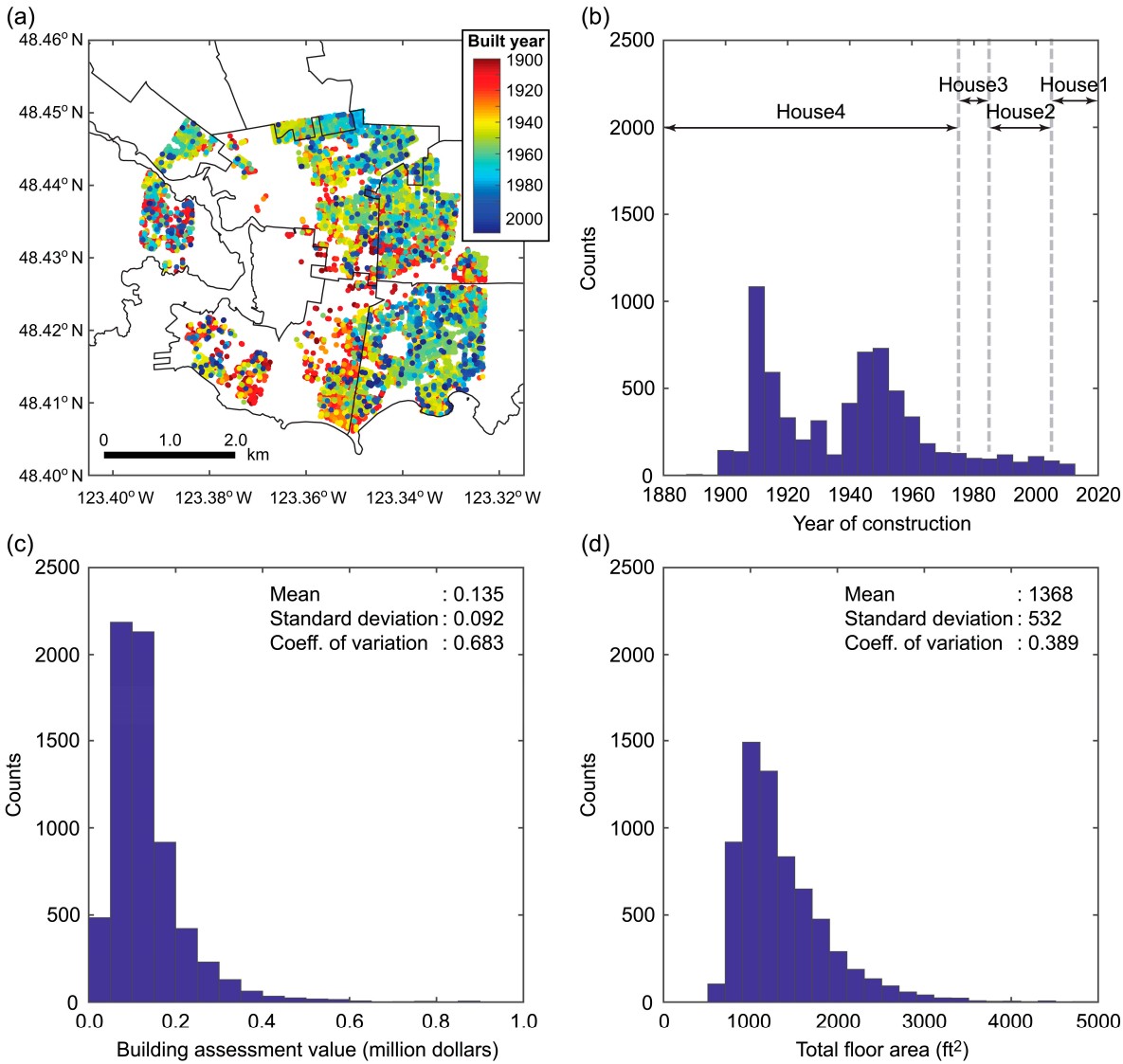

**Figure 5.** (**a**) Spatial distribution of buildings having different built years, (**b**) histogram of year of construction, (**c**) histogram of building assessment value, and (**d**) histogram of total floor area.

To provide the cost information of the selected houses, histograms of the building assessment value and total floor area are presented in Figure 5c,d, respectively. It is noted that the building assessment value, shown in Figure 5c, does not include the land assessment value and is used to represent the total repair cost of a building when it is completely damaged. This may result in the underestimation of the total repair cost, especially when some portion of the land assessment value (e.g., market values) may be affected by the earthquake damage. The total floor area data, as well as the foundation area data, indicate that the building portfolio consists of bungalows and 2-/3-story houses (note: 2-story houses are more common).

### 3.3. Seismic Fragility Model and Loss Estimation for Residential Wooden Buildings in Victoria

The seismic fragility modelling involves structural models, ground-motion record selection, and nonlinear dynamic analysis. Goda [11] developed seismic fragility functions for the four house models by considering the comprehensive regional seismic hazard information and by conducting rigorous incremental dynamic analysis [33]. The details of the seismic fragility modelling are not repeated herein, and salient information is included in the following section.

To evaluate the extent of seismic damage for a given seismic response level, four damage states are considered: slight (DS1), moderate (DS2), extensive (DS3), and collapse (DS4) (see [11] for the definitions of the damage states). The developed seismic fragility functions for Houses 1 to 4 are shown in Figure 6 (note: the model parameters of the lognormal fragility functions can be found in [9]). The seismic intensity measure for the fragility functions is SA at 0.3 s, which is representative of the fundamental vibration periods of the four house models [31]. The fragility functions for House 4, which consists of the majority of the wooden houses in Victoria, according to Figure 5b, are positioned on the left, compared with those for Houses 1 to 3, indicating that the seismic capacities of House 4 are less than the other three house models.

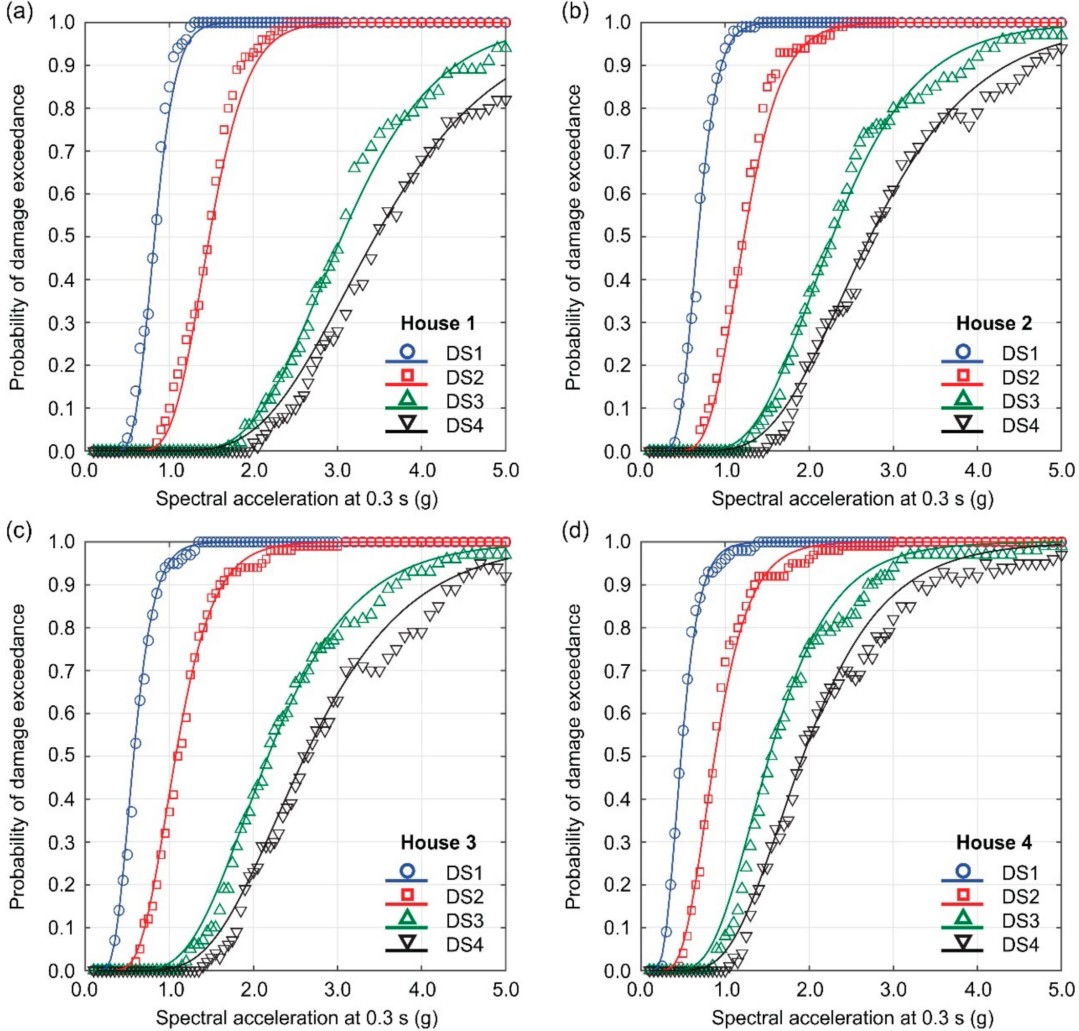

**Figure 6.** Comparison of seismic fragility functions for wooden houses in Victoria: (**a**) House 1, (**b**) House 2, (**c**) House 3, and (**d**) House 4. Symbols represent the calculated fragility estimates at individual spectral acceleration levels based on incremental dynamic analysis results, whereas curves represent the fitted functions.

To evaluate the financial seismic loss in the case of earthquake damage occurrence, the damage-loss ratio is assigned to each damage state. It is assumed that the damage–loss ratio is lognormally distributed and the mean damage–loss ratios for DS1 to DS4 are assigned as 0.05, 0.2, 0.5, and 1.0, respectively, with respect to the full replacement of the building (i.e., building assessment value shown in Figure 5c), whereas the coefficient of variations in the damage–loss ratios for DS1 to DS4 are assigned as 0.3, 0.3, 0.3, and 0.5, respectively. It is noteworthy that Goda et al. [9] demonstrated the overall consistency between the Northridge earthquake loss claim model [34] and the simulated loss results from the seismic loss model for Victoria. This comparison serves as a partial validation of the developed seismic loss model with respect to actual earthquake loss data in North America.

To carry out the seismic loss estimation for the portfolio of wooden houses, the stochastic event set (considering both the LRVF-DMF events and events from other seismic sources in southwestern British Columbia), simulated shake maps of SA at 0.3 s at the building locations and earthquake damage probabilities from the seismic fragility functions, and the calculated damage costs/losses are combined through Monte Carlo simulations. In this study, seismic losses for individual houses in the building portfolio are simulated and aggregated to the portfolio level. From the developed seismic loss model, various seismic risk outputs can be derived. As information on simulated shake maps and damage states for all stochastic events is retained, critical seismic hazard and risk maps, which are directly tied with portfolio-level seismic loss results, can also be generated [9].

## 4. Sensitivity of Fault-Source-Based Probabilistic Seismic Hazard Analysis of the Leech River Valley Fault and Devil's Mountain Fault System

The simulation-based PSHA of the LRVF-DMF system is carried out for Victoria. The results for the LRVF-DMF system are combined with those for other areal and fault sources in Southwestern British Columbia [9]. The duration of the simulation is set to 5 million years; the synthetic earthquake catalog includes circa 8.3 million simulated events above $M_w 4.8$ for sources within 400 km from Victoria (Figure 1a). The number of simulated events for the LRVF-DMF system varies, depending on the chosen models and parameters. In presenting PSHA results, three seismic intensity measures, i.e., PGA, SA(0.3 s), and SA(5.0 s), are mainly focused upon. PGA is one of the most popular measures for seismic hazard mapping purposes. The spectral acceleration (SA) at 0.3 s is a representative short-period measure and is the seismic intensity measure that is adopted to develop seismic fragility functions for typical wooden houses in southwestern British Columbia [11] (see Section 3.3), whereas SA at 5.0 s is a representative long-period seismic intensity measure, which is useful for highlighting the effects of the Cascadia subduction events in southwestern British Columbia.

As outlined in Section 3.1, various model and parameter variations are considered for the LRVF-DMF system. In Section 4.1, the effects of different spatial rupture patterns, magnitude models, and varied model parameters on magnitude–recurrence relationships are investigated. In Section 4.2, the base PSHA results for the LRVF-DMF system are discussed, in comparison with the seismic hazard results for the other seismic sources surrounding Victoria. Subsequently, sensitivity analyses of the LRVF-DMF hazard results to varied parameters of slip rate, *b* value, and $M_{max}$ shift, as well as varied logic-tree weights for the segmented versus synchronous rupture models and for the characteristic versus exponential magnitude models, are performed in Sections 4.3 and 4.4, respectively.

### 4.1. Effects of Rupture Cases and Magnitude Models on Magnitude–Recurrence Relationships

To illustrate how different magnitude models and variations of their parameters translate into different magnitude–recurrence relationships for the LRVF-DMF system, 27 magnitude–recurrence relationships (i.e., combinations of slip rate, *b* value, and $M_{max}$ shift) are shown in Figure 7 for the three rupture scenarios and for the two magnitude models. In each figure panel, the weighted average relationship, based on the 27 individual curves, is also included. The characteristic models (Figure 7a,c,e) exhibit kinks in the

magnitude recurrence curves, unlike smoothly decaying curves for the exponential models (Figure 7b,d,f). In the lower magnitude range, the earthquake occurrence frequencies for the exponential models are greater than those for the characteristic models. This trend is reversed in the high magnitude range. By inspecting the individual curves, it can be observed that the earthquake occurrence frequencies at $M_{min} = 6.0$ vary due to the values of slip rate and *b* value, whereas the maximum magnitudes attained in the curves depend on the $M_{max}$ shift. Different parameters have different degrees of influence on the magnitude–recurrence relationships and their effects also depend on the magnitude model types.

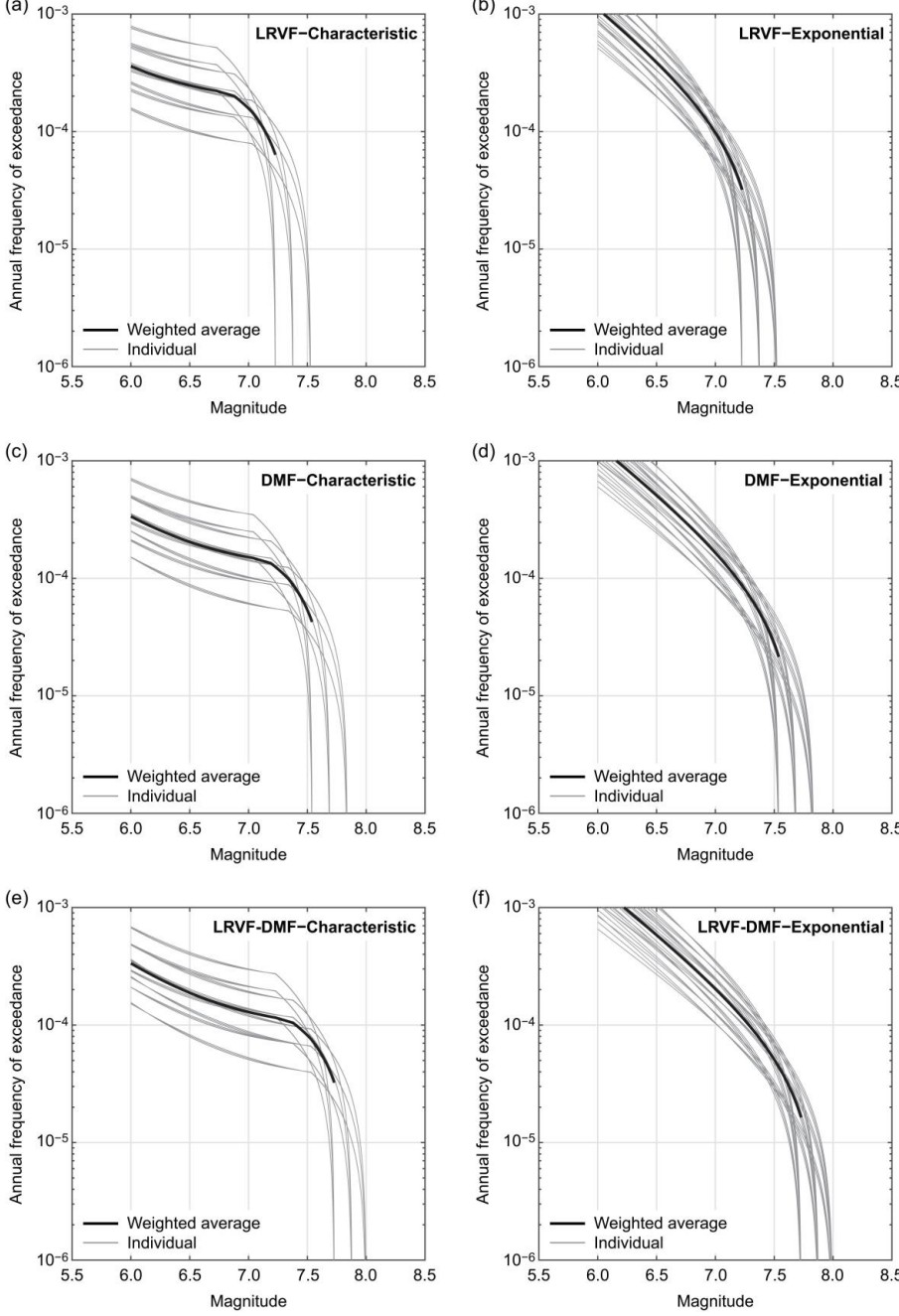

**Figure 7.** Magnitude–recurrence relationships of the LRVF-DMF system. LRVF, DMF, and LRVF-DMF (synchronous) based on the characteristic magnitude model (**a**,**c**,**e**), and LRVF, DMF, and LRVF-DMF (synchronous) based on the exponential magnitude model (**b**,**d**,**f**). In total, 27 combinations of the slip rate, *b* value, and $M_{max}$ shift are considered.

Subsequently, the seismic moment rate consistency of the different rupture scenarios and magnitude models is examined [17]. Figure 8a compares the annual seismic moment release rates based on the characteristic and the exponential models for the three earthquake rupture scenarios. The results show that the seismic moment release rates for the two magnitude models are consistent. On the other hand, Figure 8b compares the annual seismic moment release rates based on the synchronous rupture scenarios (i.e., LRVF-DMF system) and the segmented rupture scenarios (i.e., sum of the LRVF and the DMF) for the two magnitude models, for which the consistency of the seismic moment release rates for the two rupture patterns can be observed. The demonstrated consistency of the annual seismic moment rates guarantees the seismic moment rate consistency of the linear combination of any rupture scenarios and magnitude models.

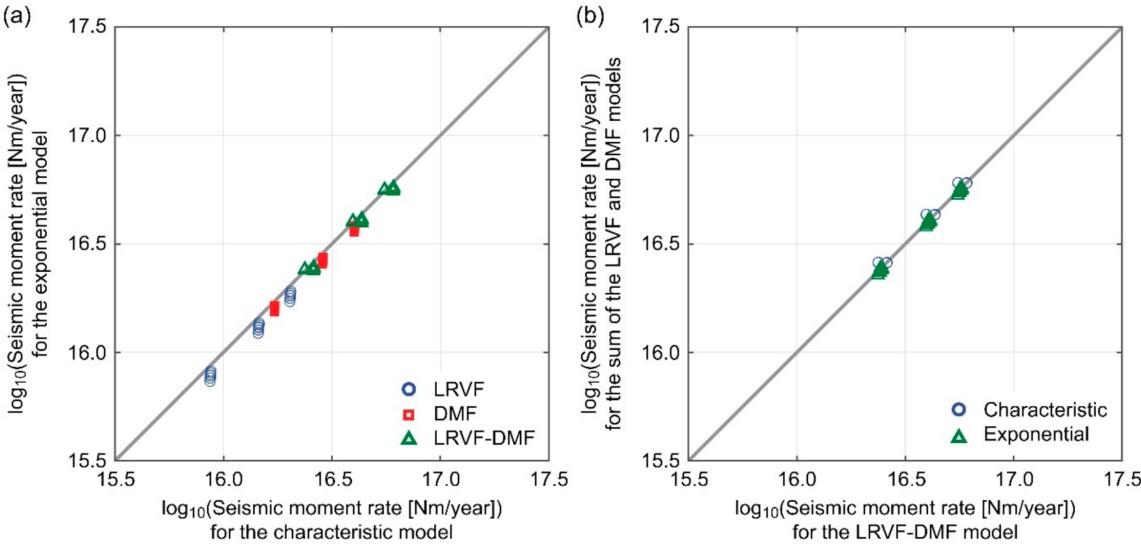

**Figure 8.** Comparison of annual seismic moment rates based on different magnitude–recurrence relationships: (**a**) characteristic versus exponential models for the LRVF, the DMF, and the LRVF-DMF system, and (**b**) LRVF-DMF system versus sum of the LRVF and the DMF for the characteristic and exponential models.

Finally, it is important to examine whether the developed magnitude–recurrence relationships shown in Figure 7 are compatible with those derived in different studies. For this purpose, we compare the weighted average relationships for the three rupture scenarios and two magnitude models (i.e., thick curves shown in Figure 7) with: (i) the two magnitude–recurrence curves for the LRVF that were derived by [16], based on the NRCan earthquake catalog, and the Li et al. microseismicity catalog, and (ii) the regional magnitude–recurrence curves for the PGT source [10]. These existing model predictions, together with the underlying data, are shown in Figure 9a; the solid and broken lines correspond to the best estimates and the lower/upper estimates, respectively. By zooming in the magnitude range of $M_\mathrm{w}5.5$ or greater, Figure 9b compares six weighted average magnitude–recurrence relationships for the LRVF-DMF system with the NRCan and Li et al. curves, as derived by [16]. From Figure 9, it can be observed that the magnitude–recurrence curves for the LRVF-DMF system lie below those for the PGT source, which ensures that the fault source seismicity is less than the regional areal seismicity, and that the fault-source-based magnitude–recurrence curves derived in this study overlap with the local earthquake-catalog-based counterparts derived by [16]. It is important to remind oneself that there are significant uncertainties associated with the varied fault-source parameters (Figure 7), which are not displayed in Figure 9b.

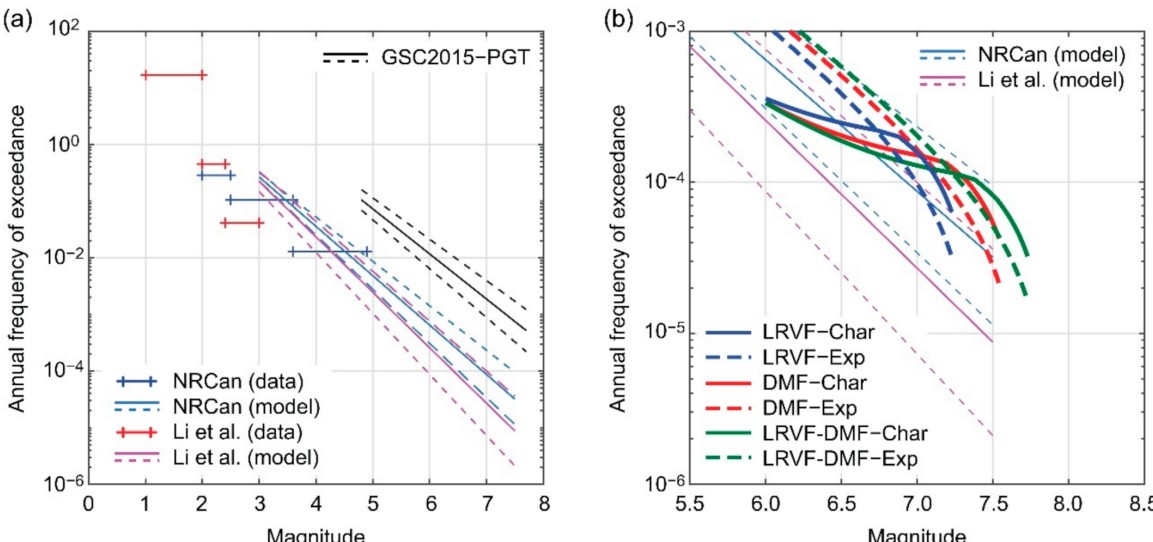

**Figure 9.** (**a**) Magnitude–recurrence relationships for the LRVF-DMF system obtained from [16] and for the PGT source zone from [10]. (**b**) Six magnitude–recurrence relationships (weighted average) for the LRVF-DMF system (three fault models and two magnitude models), as defined in Figure 7.

### 4.2. Base PSHA Results

Figure 10 shows seismic hazard curves and seismic hazard contributions, due to crustal, interface, inslab, and LRVF-DMF events for PGA, SA at 0.3 s, and SA at 5.0 s. The abbreviation letters, C, I, S, and F, stand for shallow crustal, Cascadia interface, deep inslab, and LRVF-DMF system, respectively. In the seismic hazard curve plots (Figure 10a,c,e), two combined curves are shown; the C+I+S curve is based on the result without including the LRVF-DMF system (and thus it coincides with the GSC2015 hazard model), whereas the C+I+S+F curve includes the seismic hazard contributions from the LRVF-DMF system. The differences in the two combined curves represent the effects of the LRVF-DMF system on the overall hazard results. In the seismic hazard contribution plots (Figure 10b,d,f), the relative hazard contributions are defined based on the number of stochastic events that exceed the specified hazard levels.

The seismic hazard curves and hazard contributions for different seismic intensity measures exhibit contributions from different seismic sources (i.e., C, I, S, and F). For PGA (Figure 10a,b), the inslab events (S) is the major contributor in the annual probability of exceedance range up to $6 \times 10^{-4}$ (about 1500-year return period), whereas with the decreasing annual probability of exceedance, the relative contributions from the shallow crustal earthquakes (C and F), which occur relatively close to Victoria, become dominant. The rapid increases in the seismic hazard curve and relative hazard contribution from the LRVF-DMF system in the low annual probability of exceedance range are attributed to the fact that the LRVF-DMF ruptures are rare events, as reflected in the magnitude–recurrence curves shown in Figure 7 (i.e., recurrence periods of $M_w 6.5+$ earthquakes are in the range of several thousand years). The contributions from the Cascadia interface events are not significant for PGA. The inclusion of the LRVF-DMF system in the PSHA calculations increases the combined PGA hazard curve by 6%, 10%, and 14% at the annual probability of exceedance of $2 \times 10^{-3}$, $4 \times 10^{-4}$, and $1 \times 10^{-4}$, respectively.

The results for SA at 0.3 s (Figure 10c,d) are similar to those for PGA with two notable features. One is that the transition of the dominant hazard sources from inslab to crustal events occurs at smaller annual probability of exceedance levels around $3 \times 10^{-4}$ (about 3000-year return period). The other is that the relative hazard contribution from crustal events other than the LRVF-DMF system is decreased compared with the counterpart for PGA, while the relative hazard contribution from the LRVF-DMF system remains high in the low annual probability of exceedance range. The inclusion of the LRVF-DMF system in

the PSHA calculations increases the combined SA(0.3 s) hazard curve by 6%, 10%, and 12% at the annual probability of exceedance of $2 \times 10^{-3}$, $4 \times 10^{-4}$, and $1 \times 10^{-4}$, respectively. Especially for short-period seismic intensity measures (i.e., majority of low-rise buildings), the seismic hazard contributions from the nearby LRVF-DMF system are not negligible and these events should be considered as critical earthquake scenarios for the earthquake impact assessments and disaster preparedness purposes.

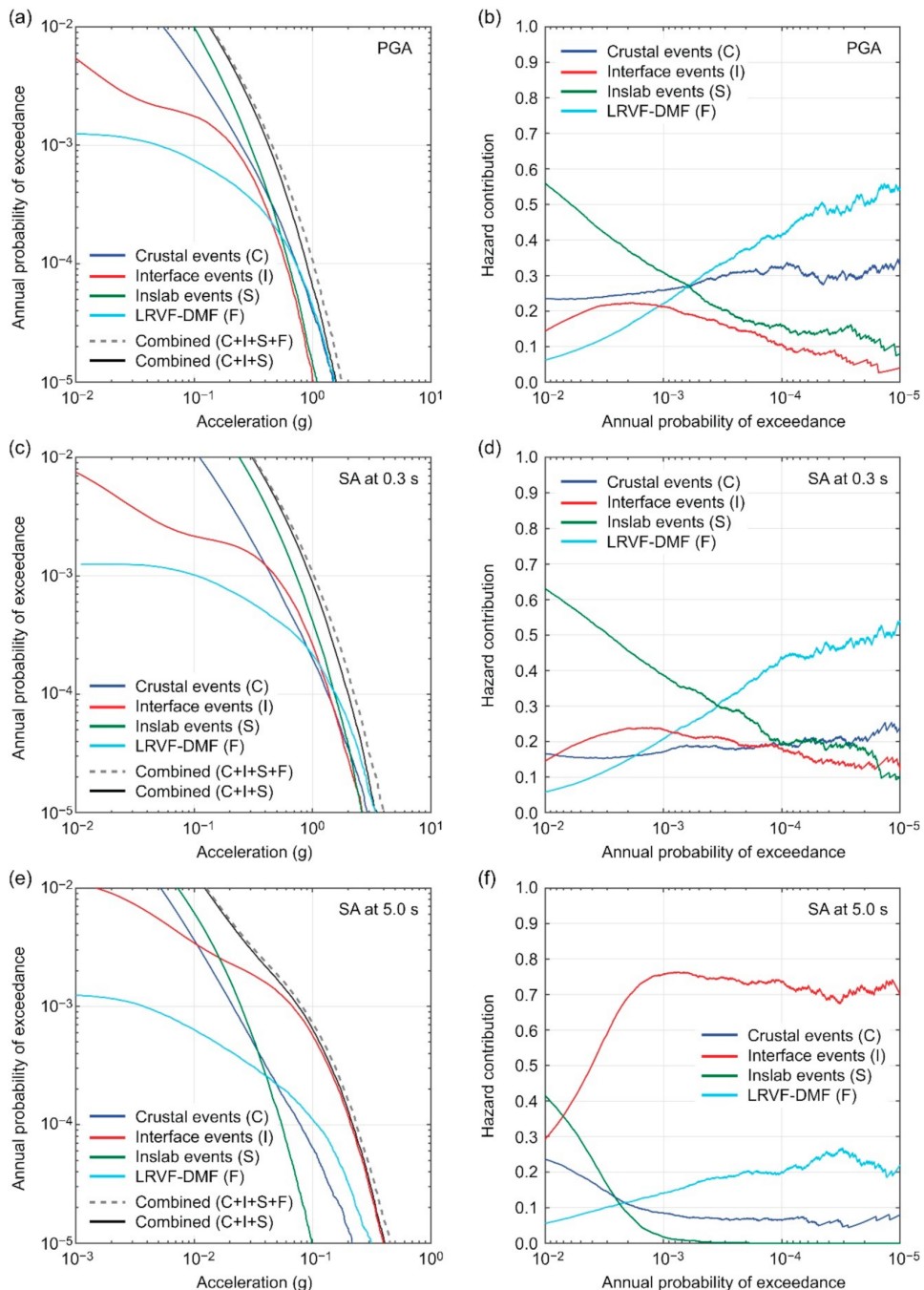

**Figure 10.** Comparison of seismic hazard curves (**a**,**c**,**e**) and seismic hazard contributions (**b**,**d**,**f**) due to crustal, interface, inslab, and LRVF-DMF events for PGA, SA at 0.3 s, and SA at 5.0 s.

The results for SA at 5.0 s (Figure 10e,f) are remarkably different from those for PGA and SA at 0.3 s, due to more significant hazard contributions from the Cascadia interface events. The dominance of this megathrust seismic source (about 70% relative contribution)

remains high for the annual probability of exceedance of $2 \times 10^{-3}$, which essentially corresponds to the typical recurrence period of the event (i.e., 450–600 years). The inclusion of the LRVF-DMF system in the PSHA calculations increases the combined SA(5.0 s) hazard curve by 8%, 8%, and 7% at the annual probability of exceedance of $2 \times 10^{-3}$, $4 \times 10^{-4}$, and $1 \times 10^{-4}$, respectively. Although the hazard contribution from the LRVF-DMF system is not the largest, it has important contributions (about 20%) at the annual probability of exceedance of $2 \times 10^{-4}$. Therefore, for long-period seismic intensity measures, this nearby fault rupture scenario may be considered after the Cascadia megathrust event.

To show the above-mentioned seismic hazard results for Victoria in a different format, uniform hazard spectra for the annual probabilities of exceedance of $4 \times 10^{-4}$ and $1 \times 10^{-4}$ (i.e., return periods of 2475 and 10,000 years) are presented in Figure 11 by distinguishing the contributing hazard sources. At the annual probability of exceedance of $4 \times 10^{-4}$, the LRVF system does not contribute significantly to the overall seismic hazard. However, at the annual probability of exceedance of $1 \times 10^{-4}$, the LRVF system becomes one of the most dominant seismic sources, especially in the short vibration period range. These results are in agreement with [15,16]; however, the main focus of our investigations is the annual probability of exceedance smaller than $4 \times 10^{-4}$ (i.e., beyond the exceedance probability for seismic hazard mapping purposes in Canada).

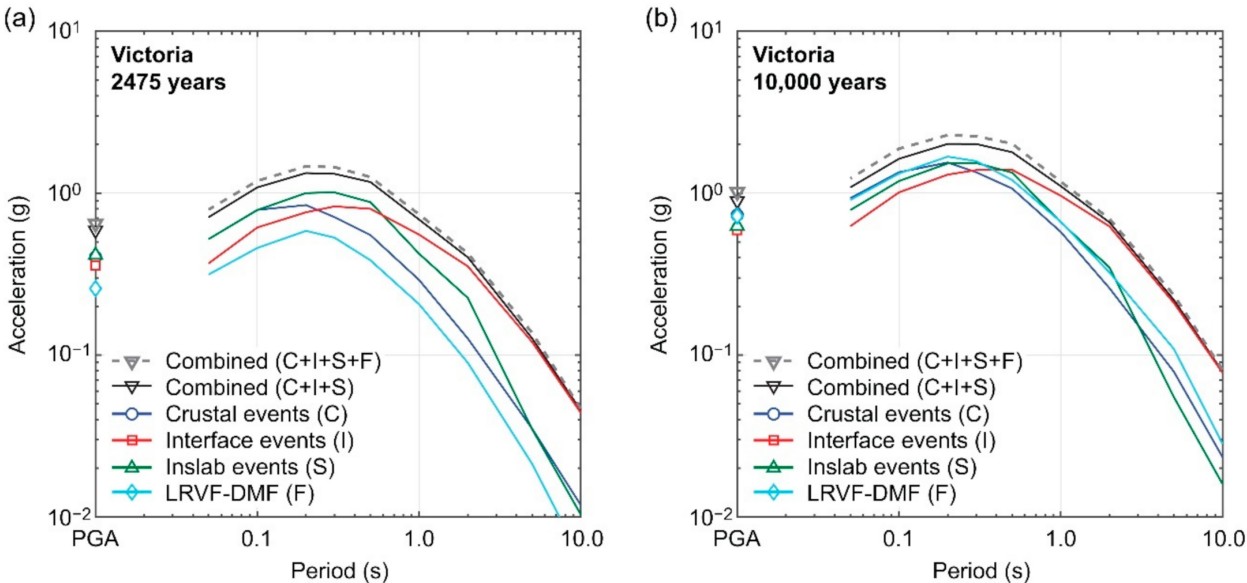

**Figure 11.** Comparison of uniform hazard spectra for Victoria by considering crustal, interface, inslab, and LRVF-DMF events: (**a**) annual probability of exceedance of $4 \times 10^{-4}$ (2475-year return period) and (**b**) annual probability of exceedance of $1 \times 10^{-4}$ (10,000-year return period).

### 4.3. Sensitivity to Slip Rate, b Value, and $M_{max}$ Shift

The base model for the LRVF-DMF system captures the 27 magnitude–recurrence relationships together with $w_1$ = 0.5 and $w_2$ = 0.5. Among the parameters of the characteristic magnitude model, some parameters are more uncertain than others and can have greater influence on the PSHA results. To assess this, 15 one-at-a-time sensitivity analyses are performed by focusing upon three key parameters: slip rate, $b$ value, and $M_{max}$ shift. For the slip rate variations, the slip rate values in the logic tree (Figure 3a) are shifted by −0.10, −0.05, 0.0, 0.05, and 0.10 mm/year; for the $b$ value variations, the $b$ values in the logic tree (Figure 3a) are shifted by −0.2, −0.1, 0.0, 0.1, and 0.2; for the $M_{max}$ shift variations, the spread of $M_{max}$ shift in the logic tree (Figure 3a) is changed to ±0.05, ±0.10, ±0.15, ±0.20 and ±0.25 (note: the central $M_{max}$ shift remains at zero). Note that $w_1$ and $w_2$ for the segmented versus synchronous rupture cases and for the characteristic versus exponential magnitude models are unaltered.

Figure 12 shows the sensitivity results for the slip rate variation. Figure 12a shows the weighted average magnitude–recurrence relationships for the five slip-rate cases. Figure 12b–d show the seismic hazard curves for PGA, SA at 0.3 s, and SA at 5.0 s, respectively, for the five slip rate cases (note: to reduce the clutter in the figure panels, only the hazard curves for the LRVF-DMF system and the combined hazard curves (i.e., C+I+S+F) are included. Figure 12e,f show uniform hazard spectra due to the LRVF-DMF system at annual probabilities of exceedance of $4 \times 10^{-4}$ and $1 \times 10^{-4}$ (2475-year and 10,000-year return periods), respectively. The increase in the slip rate leads to the increased seismic activities from the fault source and, thus, the magnitude recurrence curves shift upwards (Figure 12a). The more frequent occurrence of moderate-to-large earthquakes within the LRVF-DMF system results in the increased seismic hazard curves for the LRVF-DMF system, as well as the combined C+I+S+F cases (Figure 12b–d). For PGA, the combined seismic hazard values at the annual probability of exceedance of $1 \times 10^{-4}$ lead to a 5% increase with respect to the combined C+I+S case when the mean slip rate is 0.15 mm/year. A 20% increase is achieved when the mean slip rate is increased to 0.35 mm/year. On the other hand, the effects of the slip rate variation are less noticeable when longer-period seismic intensity measures, such as SA(5.0 s), are considered (Figure 12d). In terms of uniform hazard spectra for the LRVF-DMF system (Figure 12e,f), the effects of the slip rate increase are more pronounced at the annual probability of exceedance of $4 \times 10^{-4}$ than at the annual probability of exceedance of $1 \times 10^{-4}$ because the changes in the slip rate directly affect the occurrence frequency of the LRVF-DMF events (Figure 12a).

Figure 13 shows the sensitivity results for the *b* value variation. The configurations of the presented results are the same as in Figure 12. As shown in Figure 13a, steepening the slope of the magnitude–recurrence relationships has more influence on the small-to-moderate magnitude range (e.g., $M_w6$ to $M_w6.5$). These changes shift the seismic moment release in one magnitude range to another but do not alter the total seismic moment release from the fault source. The effects of the *b* value variation are clearly seen in the seismic hazard curves for the LRVF-DMF system, but the effects on the combined hazard curves are not pronounced (Figure 13b–d). Nearly all cases of the combined hazard curves overlap with the base case, especially at the smaller annual probability of exceedance levels. The same observations can be obtained by inspecting the uniform hazard spectra for the LRVF-DMF system at the two annual probabilities of exceedance (Figure 13e,f).

Figure 14 shows the sensitivity results for the $M_{max}$ shift spread variation, with the same configurations of the presented results as Figure 12. The weighted average magnitude–recurrence relationships exhibit small differences in the large magnitude range (Figure 14a). Because of the smaller effects of the $M_{max}$ shift spread variation, we hardly observe any noticeable effects on the seismic hazard curves (Figure 14b–d) nor on the uniform hazard spectra for the LRVF-DMF system (Figure 14e,f).

Overall, among the three examined parameters, the influence of the slip rate is the most significant and thus the uncertainty characterization of this parameter needs to be scrutinized further in conducting a fault-source-based PSHA. This is expected because the slip rate changes the occurrence frequency of seismic events from the fault deformation zone, while the *b* value variation and the $M_{max}$ shift spread variation, as examined herein, do not change the total seismic moment release.

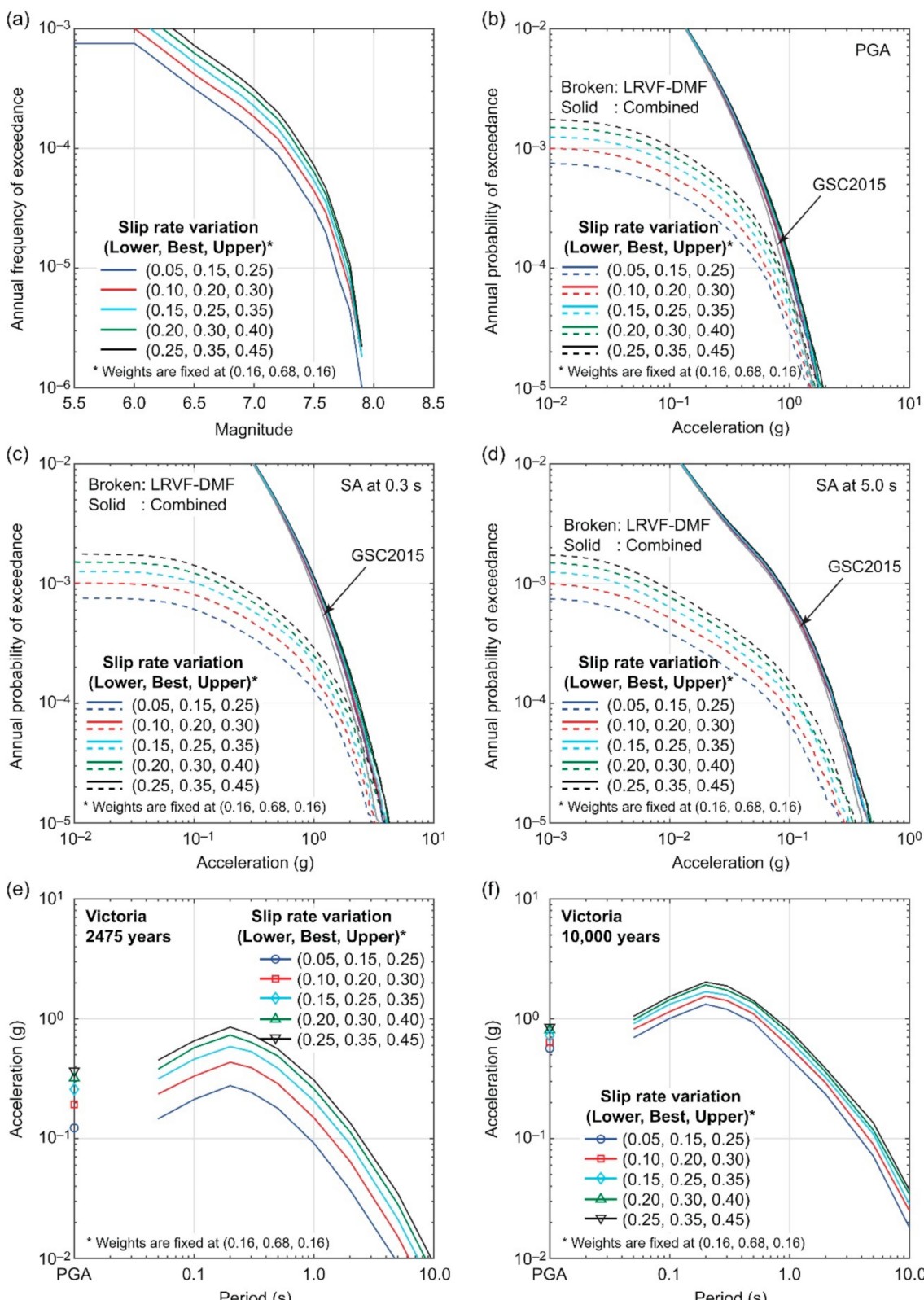

**Figure 12.** Sensitivity to slip rate variation: (**a**) magnitude–recurrence relationships, (**b**–**d**) seismic hazard curves for PGA, SA at 0.3 s, and SA at 5.0 s, and (**e**,**f**) uniform hazard spectra for the LRVF-DMF system at annual probabilities of exceedance of $4 \times 10^{-4}$ and $1 \times 10^{-4}$ (2475-year and 10,000-year return periods).

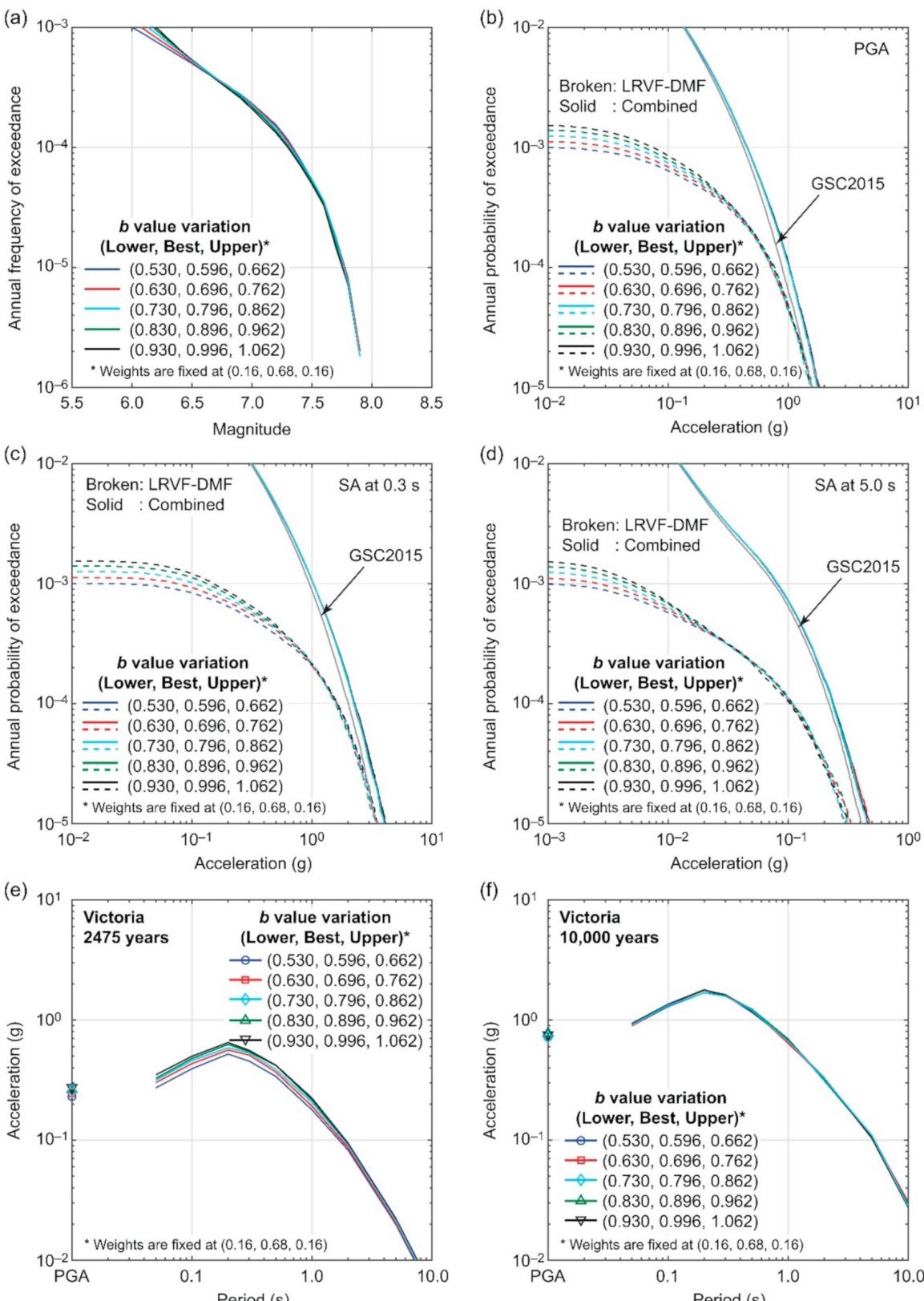

**Figure 13.** Sensitivity to *b* value variation: (**a**) magnitude–recurrence relationships, (**b**–**d**) seismic hazard curves for PGA, SA at 0.3 s, and SA at 5.0 s, and (**e**,**f**) uniform hazard spectra for the LRVF-DMF system at annual probabilities of exceedance of $4 \times 10^{-4}$ and $1 \times 10^{-4}$ (2475-year and 10,000-year return periods).

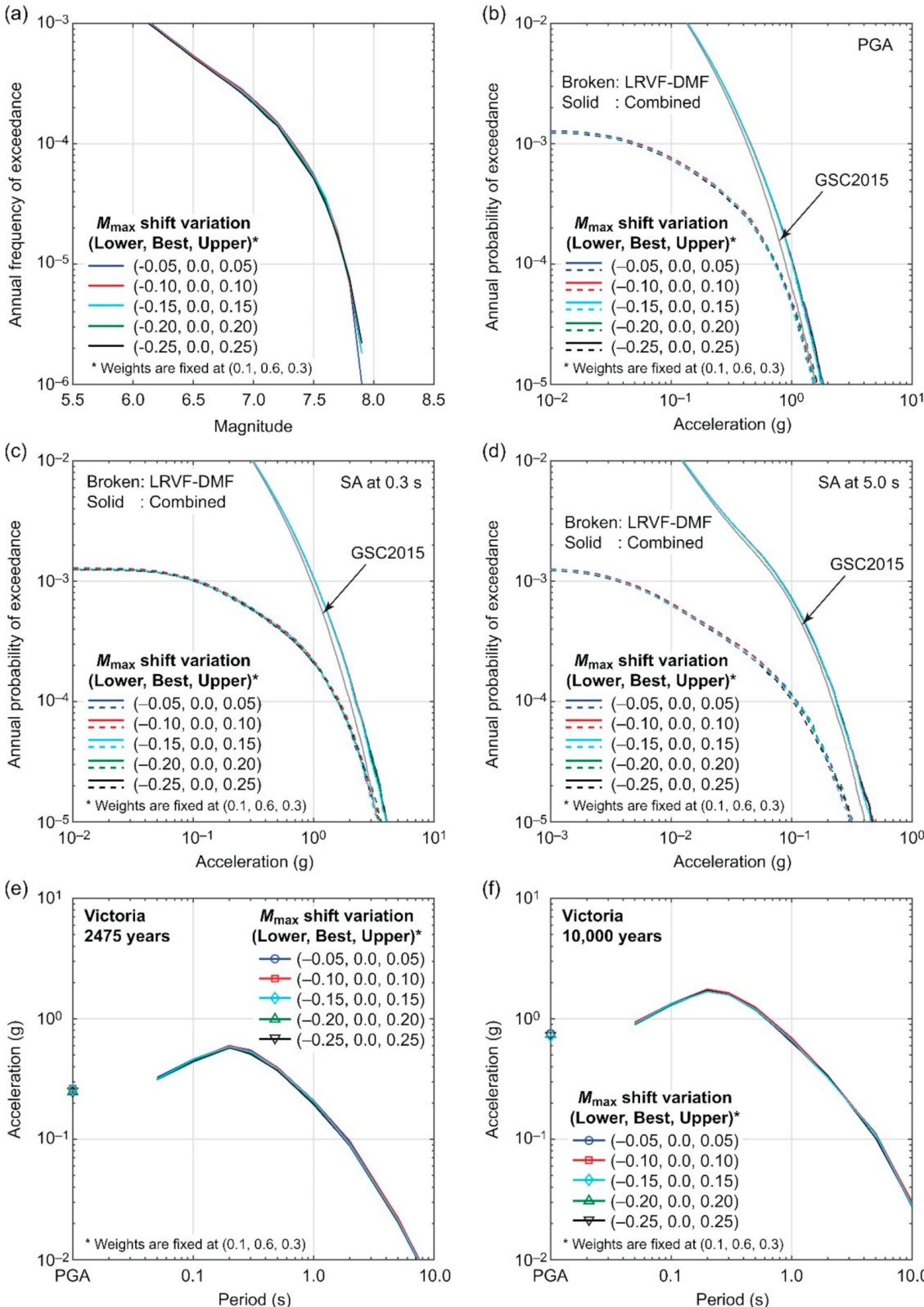

**Figure 14.** Sensitivity to $M_{max}$ shift variation: (**a**) magnitude–recurrence relationships, (**b**–**d**) seismic hazard curves for PGA, SA at 0.3 s, and SA at 5.0 s, and (**e**,**f**) uniform hazard spectra for the LRVF-DMF system at annual probabilities of exceedance of $4 \times 10^{-4}$ and $1 \times 10^{-4}$ (2475-year and 10,000-year return periods).

*4.4. Sensitivity to Logic-Tree Weight Variations for Segmented Versus Synchronous Rupture Scenarios and Characteristic Versus Exponential Magnitude Models*

The rupture scenarios and magnitude models are highly uncertain, and it is not an easy task to determine these model components solely based on available geological data. Recognizing the major epistemic uncertainty, we conduct sensitivity analyses related to these components by varying the logic-tree weights $w_1$ and $w_2$ (Figure 3b). Three types of investigations are carried out. For the first two cases, either values of $w_1$ only or $w_2$ only are varied from 0.0 to 1.0, with a 0.2 increment by holding the other weight at a default value of 0.5, whereas for the third case, both $w_1$ and $w_2$ are changed and all 36 combinations of $w_1$ and $w_2$ (with 0.2 increment) are considered. The logic-tree branches for the recurrence parameters and the ground-motion models are the same as the base case (see Figure 3). It is noteworthy that the aim of the sensitivity analyses presented herein is to quantify the variability of the seismic hazard estimates, rather than deriving the most accurate seismic hazard estimates.

Figure 15 shows the sensitivity results for the logic-tree weight variation of the segmented versus synchronous rupture scenarios; the configurations of the presented results are the same as in Figure 12. A smaller value of $w_1$ leads to more synchronous ruptures of the LRVF-DMF source and thus tends to shift the seismic moment release towards the larger magnitude range. On the other hand, a larger value of $w_1$ treats the LRVF and the DMF as independent fault sources; therefore, the maximum magnitude is restricted to smaller values (due to shorter fault lengths) but with more frequent occurrences of moderate earthquakes from the two sources. Such different rupture behavior can be seen in Figure 15a. When the effects of different rupture scenarios are propagated to the seismic hazard curves (Figure 15b–d) and the uniform hazard spectra (Figure 15e,f), their influences are noticeable for the LRVF-DMF hazard curves and uniform hazard spectra at an annual probability of exceedance of $4 \times 10^{-4}$. In contrast, their influences on the combined hazard curves and the LRVF-DMF uniform hazard spectra at annual probability of exceedance of $1 \times 10^{-3}$ are similar to the base case.

Figure 16 shows the sensitivity results for the logic-tree weight variation of the characteristic versus exponential magnitude models; the configurations of the presented results are the same as in Figure 12. A smaller value of $w_2$ places a greater emphasis on the Gutenberg–Richter-type magnitude–recurrence relationship and thus results in more frequent occurrences of small-to-moderate events in comparison with large events. A larger value of $w_2$ favors the occurrence of larger events, although the occurrence of such large events is less frequent. Regarding how the effects of different magnitude–recurrence relationships are propagated to the seismic hazard curves and uniform hazard spectra, general observations that are made for Figure 15 are applicable to Figure 16. By comparing the results for Figures 15 and 16, it can be observed that the logic-tree weight variation of the characteristic versus exponential magnitude models has more influence than the logic-tree weight variation of the segmented versus synchronous rupture scenarios.

Figure 17 presents a more comprehensive sensitivity analysis by simultaneously varying the weights (note: the same configurations of the presented results as Figure 12). The ranges of the varied magnitude–recurrence curves, seismic hazard curves, and uniform hazard spectra are wider than those in Figures 15 and 16 (as expected). The results presented in Figure 17 highlight that the seismic hazard contributions from the LRVF-DMF system depend on the annual probability of exceedance level, especially for the annual probability of exceedance of $4 \times 10^{-4}$ (Figure 17e). Overall, the sensitivity analysis results presented in this section demonstrate the substantial variability of the seismic hazard results, due to the fault scenarios and the magnitude models.

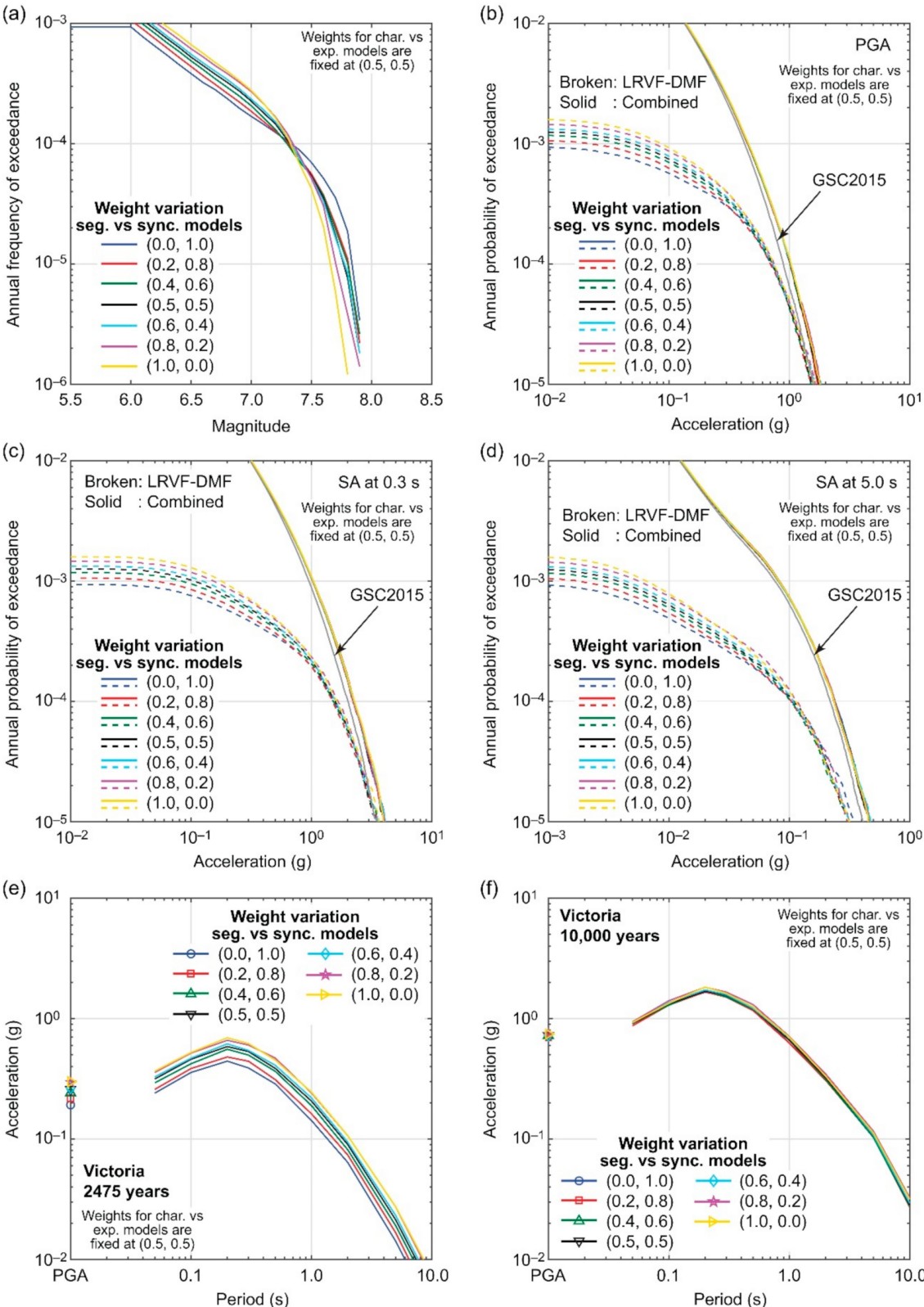

**Figure 15.** Sensitivity to logic-tree weight variations of the segmented versus synchronous rupture models (note: logic-tree weights for the characteristic versus exponential magnitude models are set to 0.5): (**a**) magnitude–recurrence relationships, (**b**–**d**) seismic hazard curves for PGA, SA at 0.3 s, and SA at 5.0 s, and (**e**,**f**) uniform hazard spectra for the LRVF-DMF system at annual probabilities of exceedance of $4 \times 10^{-4}$ and $1 \times 10^{-4}$ (2475-year and 10,000-year return periods).

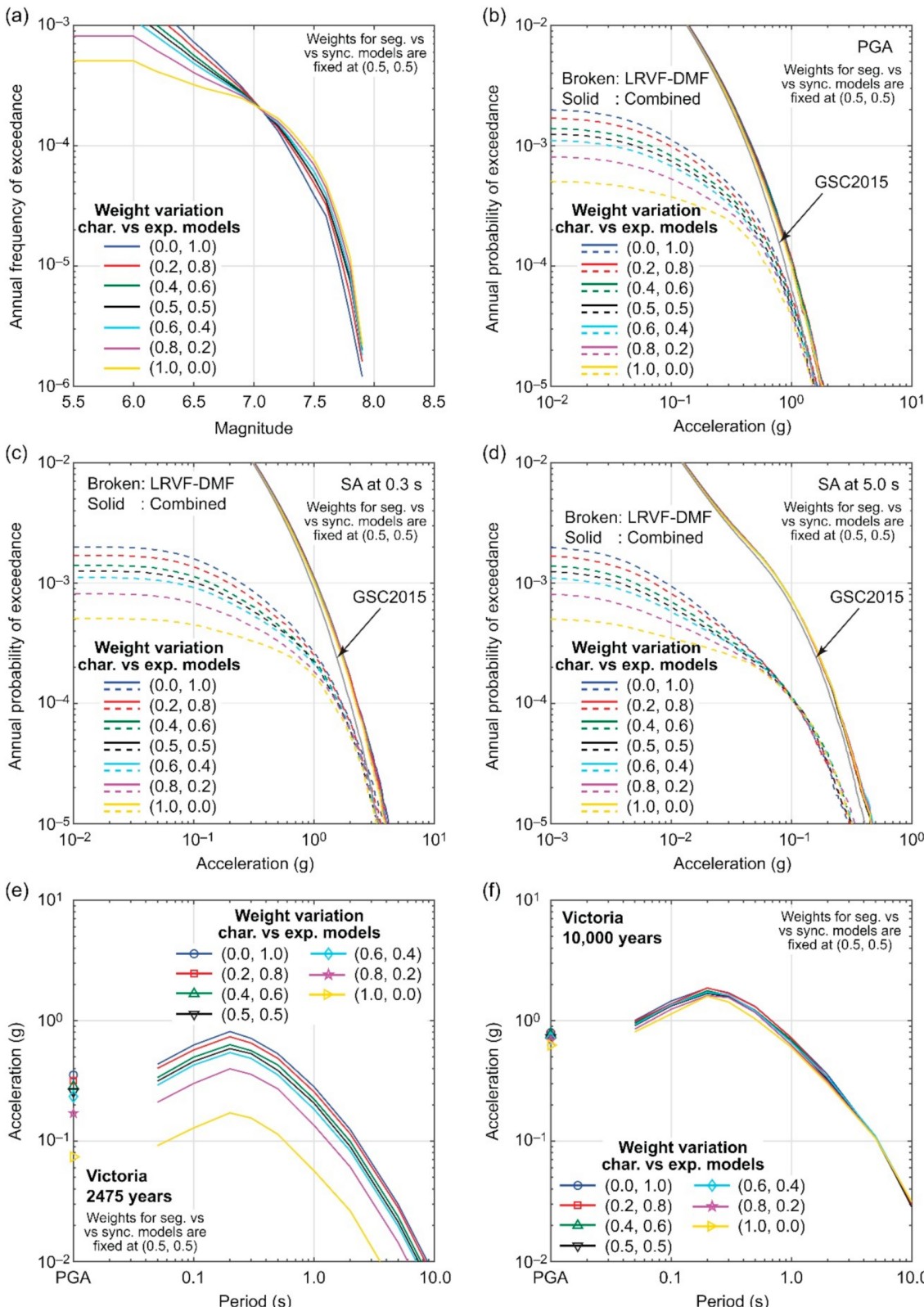

**Figure 16.** Sensitivity to logic-tree weight variations for the characteristic versus exponential magnitude models (note: logic-tree weights of the segmented versus synchronous rupture models are set to 0.5): (**a**) magnitude–recurrence relationships, (**b**–**d**) seismic hazard curves for PGA, SA at 0.3 s, and SA at 5.0 s, and (**e**,**f**) uniform hazard spectra for the LRVF-DMF system at annual probabilities of exceedance of $4 \times 10^{-4}$ and $1 \times 10^{-4}$ (2475-year and 10,000-year return periods).

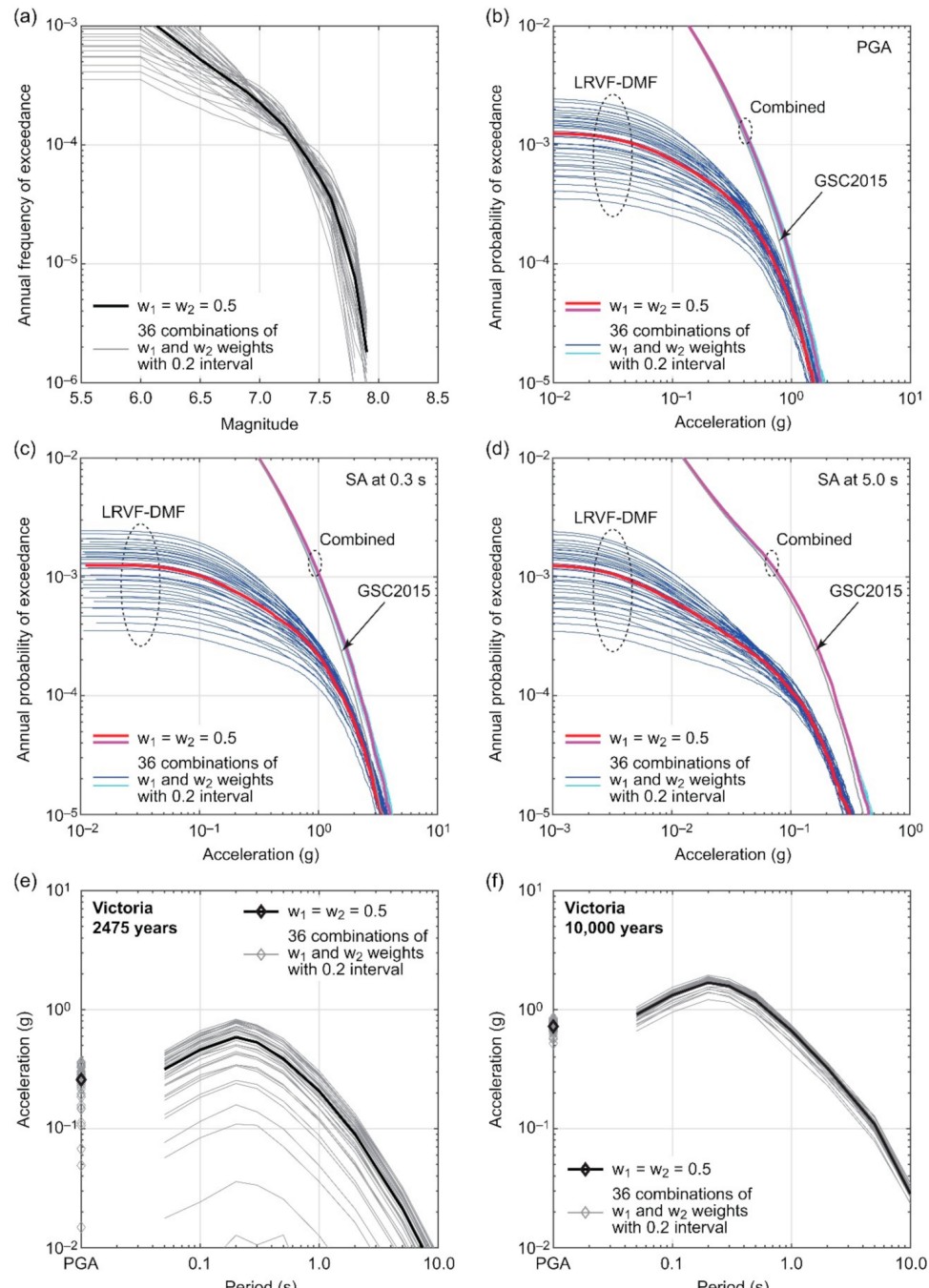

**Figure 17.** Sensitivity to simultaneous logic-tree weight variations for the segmented versus synchronous rupture models and for the characteristic versus exponential magnitude: (**a**) magnitude–recurrence relationships, (**b–d**) seismic hazard curves for PGA, SA at 0.3 s, and SA at 5.0 s, and (**e,f**) uniform hazard spectra for the LRVF-DMF system at annual probabilities of exceedance of $4 \times 10^{-4}$ and $1 \times 10^{-4}$ (2475-year and 10,000-year return periods). In panels (**b–d**), dotted circles are shown to distinguish two sets of seismic hazard curves for the LRVF-DMF system only and for the combined case.

## 5. Sensitivity of Fault-Source-Based Probabilistic Seismic Risk Analysis of the Leech River Valley Fault and Devil's Mountain Fault System

The fault-source-based seismic hazard model for the LRVF-DMF system is integrated with the exposure and seismic fragility models for the residential wooden houses in Victoria. In Section 5.1, the seismic loss results for the base case (Section 4.2) are discussed by investigating the effects of the LRVF-DMF system on the portfolio-level seismic loss curves and risk-based critical earthquake scenario maps for Victoria. Building upon the

base case results, the sensitivity analyses of the portfolio-level seismic loss curves to the varied parameters of slip rate, $b$ value, and $M_{max}$ shift, as well as varied logic-tree weights for the segmented versus synchronous rupture models, and for the characteristic versus exponential magnitude models, are performed in Sections 5.2 and 5.3, respectively.

### 5.1. Base Case

The standard outputs from the earthquake loss model, such as seismic loss curve and seismic loss disaggregation, are useful for understanding the extent of regional seismic loss at stake. Figure 18a shows seismic loss curves for the building portfolio that is developed for the residential housing stock in Victoria by distinguishing the contributing seismic sources (i.e., C, I, S, and F). The differences of the two combined curves (i.e., C+I+S versus C+I+S+F) indicate the seismic loss contributions by the LRVF-DMF system with respect to all other sources surrounding Victoria. As observed for the seismic hazard curves in Section 4.2, the LRVF-DMF system significantly contributes to the overall portfolio seismic loss, particularly in the low annual probability of exceedance range. The increased earthquake impact due to the inclusion of the LRVF-DMF system is large; at the annual probabilities of exceedance of $2 \times 10^{-3}$, $4 \times 10^{-4}$, and $1 \times 10^{-4}$, the combined seismic loss values, which are equivalent to a popular financial risk metric, value-at-risk (VaR), are increased by 15%, 23%, and 20%, respectively. To examine the relative loss contributions from different seismic sources, Figure 18b presents the seismic loss disaggregation plot, where the relative loss contributions are defined based on the number of stochastic events that exceed the specified loss levels. The results clearly show that, with the decrease in the annual probability of exceedance (i.e., higher return period), the relative contributions from crustal events (C and F) become dominant, and at the annual probability of exceedance of $5 \times 10^{-4}$ (i.e., 2000-year return period) or lower, the LRVF-DMF system is the most dominant.

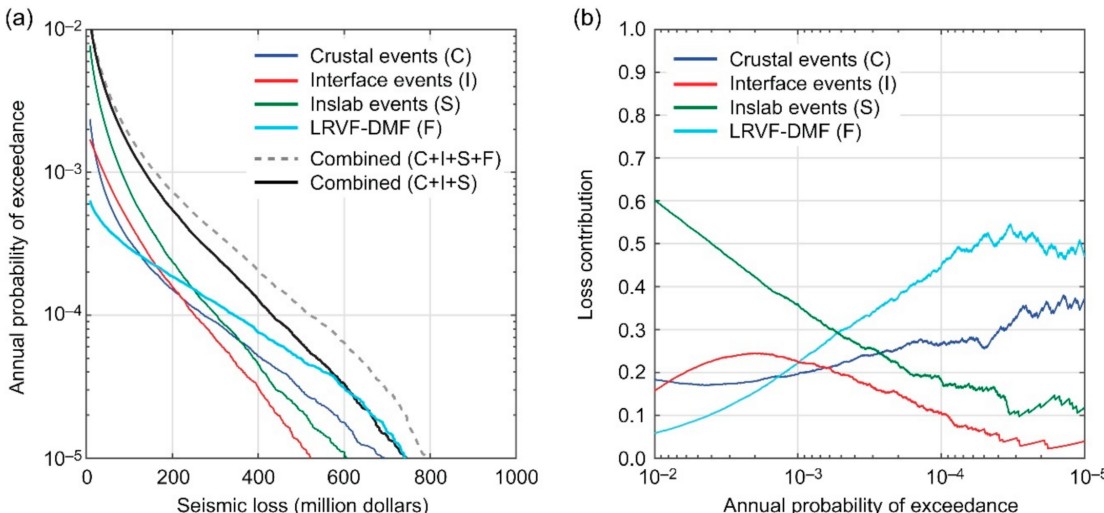

**Figure 18.** Comparison of exceedance probability curves of portfolio seismic loss (**a**) and seismic loss contributions (**b**) due to crustal, interface, inslab, and LRVF-DMF events.

To provide further insights into the contributing seismic loss events, geographical seismic loss disaggregation is performed by extracting information of seismic events that contribute to the specified loss levels. The results for all loss events, events that result in the portfolio loss at the annual probability of exceedance of $4 \times 10^{-4}$ or smaller, and events that result in the portfolio loss at the annual probability of exceedance of $1 \times 10^{-4}$ or smaller, are shown in Figure 19. Figure 19a shows the overall spatial distribution of seismic loss events; various sources of events result in the seismic loss, including the shallow crustal events, off-shore Cascadia subduction zone, inslab source in Puget Sound, and the LRVF-DMF system. With the increase of the seismic loss threshold, smaller

magnitude and more distant events are eliminated, and the loss contributions from the nearby shallow crustal sources become dominant (Figure 19b,c). According to the results shown in Figures 18 and 19, different earthquakes, occurring at different locations and depths, and with different magnitudes, can lead to similar levels of portfolio seismic loss. Therefore, for regional seismic risk management purposes, multiple critical earthquake scenarios should be selected by reflecting different risk levels and different earthquake sources, rather than the ad hoc selection of one or two scenarios.

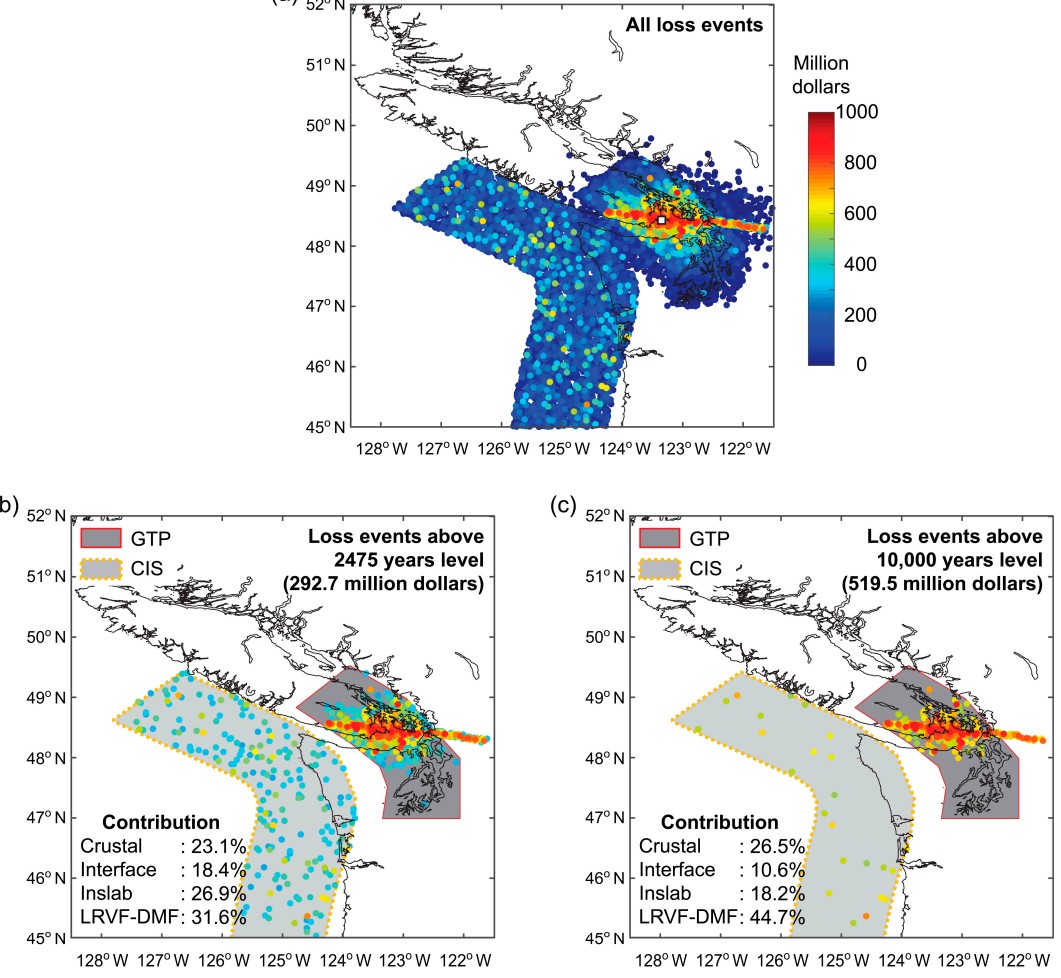

**Figure 19.** Spatial distributions of seismic events contributing to different seismic loss levels: (**a**) all loss events, (**b**) annual probability of exceedance of $4 \times 10^{-4}$ (2475-year return period), and (**c**) annual probability of exceedance of $1 \times 10^{-4}$ (10,000-year return period). All events are represented as point sources. GTP corresponds to a seismic source zone for deep inslab events, whereas CIS corresponds to a seismic source zone for the Cascadia subduction interface events.

Lastly, the integrated use of the outputs from the developed seismic loss model facilitates the risk-based identification of critical earthquake scenarios, which are useful for different stakeholders for earthquake risk management purposes [9]. To demonstrate this, seismic shake maps, damage state maps, and seismic loss ratio maps for the building portfolio in Victoria for the annual probabilities of exceedance of $4 \times 10^{-4}$ and $1 \times 10^{-4}$ (2475-year and 10,000-year return periods) are shown in Figure 20. The seismic shake maps shown in Figure 20a,b display the spatial variations of shaking intensities at building locations. Due to the spatially correlated variability and local site conditions, values of SA at 0.3 s change gradually, and for different events, hot spots where experienced ground motions are greater than the surrounding areas may appear at different locations. Damage state maps are more direct building-level risk outputs, which are shown in Figure 20c,d for the annual probabilities of exceedance of $4 \times 10^{-4}$ and $1 \times 10^{-4}$. The damage state maps

exhibit possible patterns of city-wide building damage distribution. Based on the detailed information on individual buildings, supplementary city-level building damage statistics, such as the numbers of houses with DS0, DS1, DS2, DS3, and DS4, can be obtained (see the figures). This form of seismic risk outputs may be more useful to emergency risk managers and structural engineers who may need to evaluate the actual extent of structural damage and to develop effective building inspection and repair procedures. Similarly, seismic risk maps that are based on the calculated loss ratios for individual houses, can be produced (Figure 20e,f). The seismic loss ratio maps are likely to be more useful for policy makers and insurers/reinsurers to understand the degree of financial seismic risk and their impact to the regional economy. The building-level and city-wide hazard and risk maps displayed in Figure 20 are directly related and, thus, for the same annual probability of exceedance level, the spatial patterns of the hazard and risk metrics match, and can be linked back to the portfolio-level risk outputs (Figures 18 and 19). Such integrated use of hazard and risk outputs from advanced earthquake loss models should be promoted across various sectors and to stakeholders who are concerned with catastrophic earthquake risks.

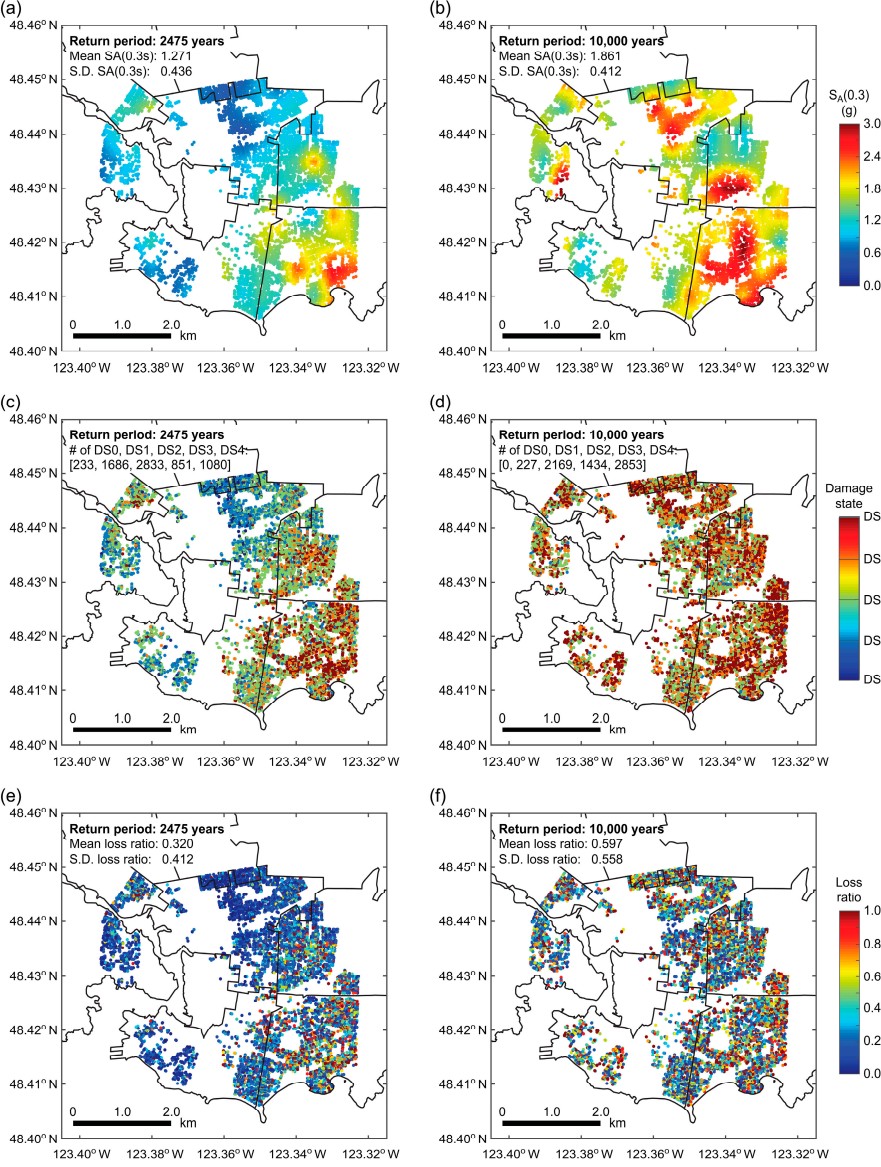

**Figure 20.** Critical earthquake scenarios in terms of seismic shake maps (**a**,**b**), damage state maps (**c**,**d**), and seismic loss ratio maps (**e**,**f**) for annual probabilities of exceedance of $4 \times 10^{-4}$ and $1 \times 10^{-4}$ (2475-year and 10,000-year return periods).

## 5.2. Sensitivity to Slip Rate, b Value, and $M_{max}$ Shift

As explored in Section 4.3, some of the key parameters related to the LRVF-DMF system are uncertain, and they can have significant influences on the seismic hazard results. We investigate their effects on the portfolio-level seismic risk results through one-at-a-time sensitivity analysis. The set-up of the sensitivity analysis is similar to that outlined in Section 4.3 by varying the three parameters: slip rate, *b* value, and $M_{max}$ shift. The two logic-tree weights, $w_1$ and $w_2$, are maintained at the default values of 0.5.

Figure 21a,c,e show the sensitivity results of the portfolio seismic loss curve for Victoria to slip rate, *b* value, and $M_{max}$ shift variations. To show the increased earthquake impact due to the LRVF-DMF system, ratios of portfolio seismic loss based on the combined GSC2015 and LRVF-DMF models to portfolio seismic loss based on the GSC2015 model are shown in Figure 21b,d,f. Figure 21a,b highlight significant effects due to the changes in the slip rate. When the slip rate parameters for the best branch are changed from 0.15 to 0.35 mm/year, the portfolio seismic loss values at the annual probabilities of exceedance of $2 \times 10^{-3}$, $4 \times 10^{-4}$, and $1 \times 10^{-4}$ are increased by 12%, 17%, and 12%, respectively (i.e., from 1.09 to 1.21, from 1.15 to 1.32, and from 1.16 to 1.28, respectively; see Figure 21b). On the other hand, the effects of the *b* value and $M_{max}$ shift variations are insignificant. These results are broadly consistent with those shown in Section 4.3.

## 5.3. Sensitivity to Logic-Tree Weight Variations for Segmented Versus Synchronous Rupture Models and Characteristic Versus Exponential Magnitude Models

We investigate the sensitivity of the portfolio-level seismic risk to the logic-tree weight variations for the segmented versus synchronous rupture models and for the characteristic versus exponential magnitude models. The set-up for this loss sensitivity analysis is similar to the hazard sensitivity analysis conducted in Section 4.4. The values of the logic-tree weights $w_1$ and $w_2$ are varied between 0.0 and 1.0 with 0.2 increment either individually or simultaneously.

Figure 22 presents the sensitivity results of the portfolio seismic loss for Victoria to individual and simultaneous logic-tree weight variations for the segmented versus synchronous rupture models and for the characteristic versus exponential magnitude. Figure 22a,c,e show the results in terms of exceedance probability loss curve, whereas Figure 22b,d,f show the results in terms of portfolio seismic loss ratio between the combined GSC2015 and LRVF-DMF models and the GSC2015 model. Figure 22a,b indicate that the effects due to the weight variations for the rupture scenarios are not particularly significant, noting that moderate degrees of the loss variability can be seen in the annual probability of exceedance range between $1 \times 10^{-3}$ and $2 \times 10^{-4}$. On the other hand, the effects due to the weight variations for the magnitude models are more pronounced, as shown in Figure 22c,d. These observations agree with those made in Section 4.4. When the effects of both weight variations are combined (Figure 22e,f), the exceedance probability curves, as well as the seismic loss ratios, exhibit significant variability, and some extreme combinations of the rupture scenarios and magnitude models can result in the exceedance of the GSC2015 loss curve by the LRVF-DMF loss curve alone at the annual probability of exceedance of $1 \times 10^{-4}$. The seismic loss sensitivity results highlight the importance of the consideration of the critical nearby fault source to an urban area and its uncertainty characterization.

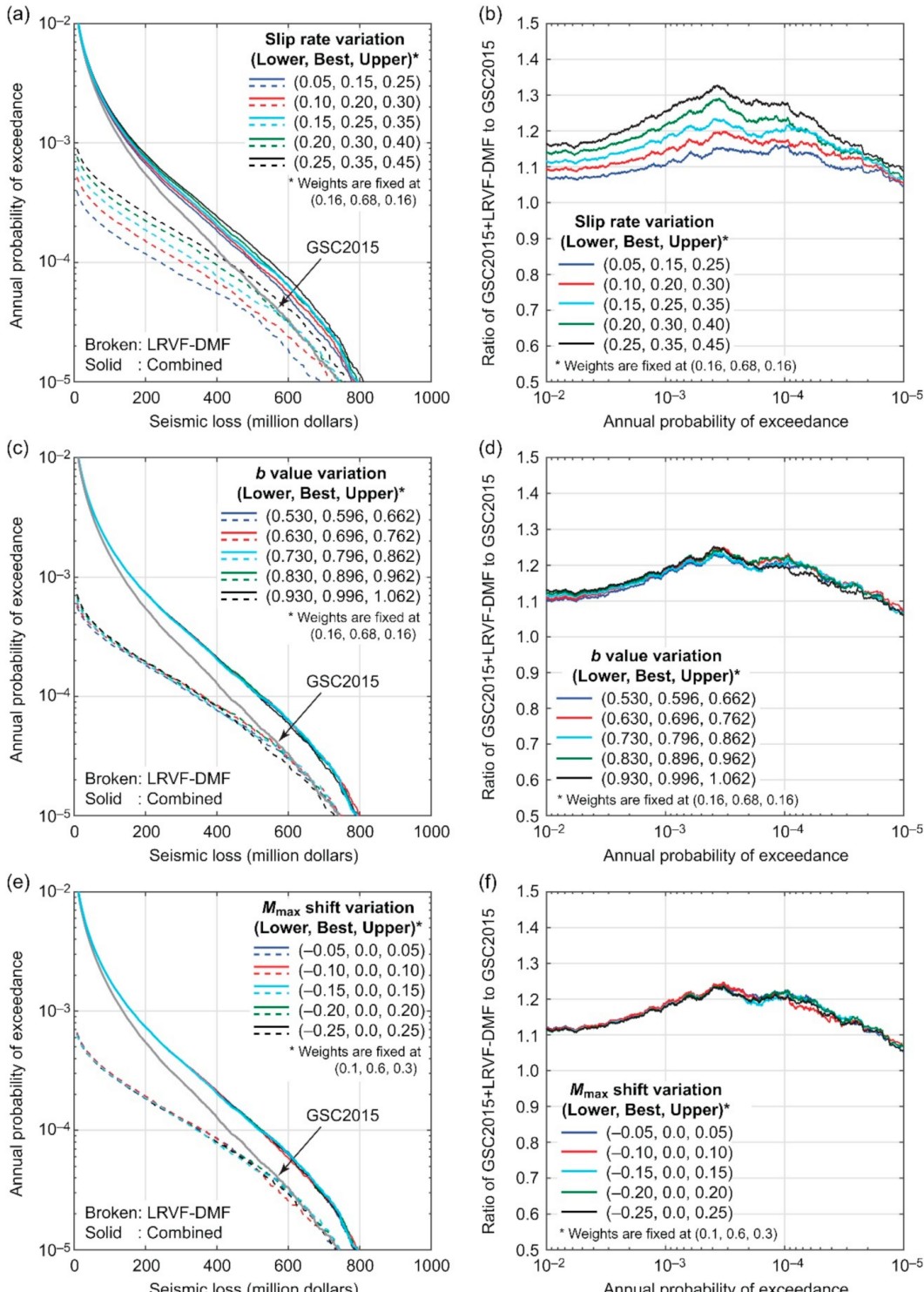

**Figure 21.** Sensitivity of exceedance probability curve of portfolio seismic loss to slip rate, *b* value, and $M_{max}$ shift variations (**a**,**c**,**e**). Ratios of portfolio seismic loss based on the combined GSC2015 and LRVF-DMF models to portfolio seismic loss based on the GSC2015 model due to slip rate, *b* value, and $M_{max}$ shift variations (**b**,**d**,**f**).

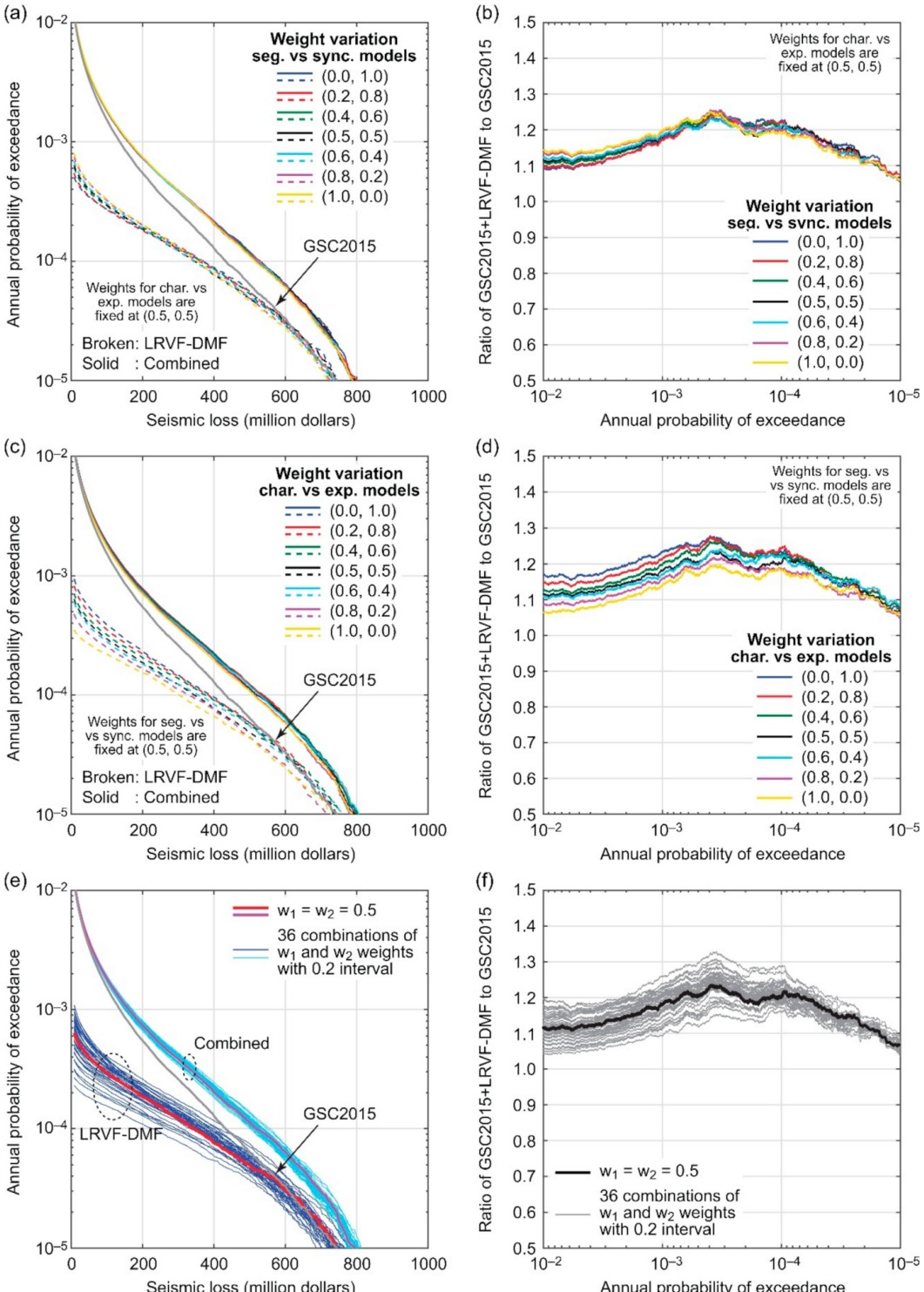

**Figure 22.** Sensitivity of exceedance probability curve of portfolio seismic loss to individual and simultaneous logic-tree weight variations for the segmented versus synchronous rupture models and for the characteristic versus exponential magnitude (**a,c,e**). Ratios of portfolio seismic loss based on the combined GSC2015 and LRVF-DMF models to portfolio seismic loss based on the GSC2015 model due to individual and simultaneous logic-tree weight variations for the segmented versus synchronous rupture models and for the characteristic versus exponential magnitude (**b,d,f**). In panels (**b–d**), dotted circles are shown to distinguish two sets of seismic hazard curves for the LRVF-DMF system only and for the combined case.

## 6. Conclusions

Characterizing the earthquake rupture of an active fault source and quantifying its seismic hazard potential are challenging due to insufficient geological and geophysical information. When the fault source is close to an urban area, despite the large uncertainty we face, its earthquake impact needs to be evaluated as accurately as possible. The Leech River Valley Fault (LRVF) and the Devil's Mountain Fault (DMF) in southern Vancouver Island, British Columbia, Canada, which can be regarded as a fault system and thus could rupture synchronously, are an exemplar of such situations. These faults pass underneath the City of Victoria and can pose significant risks to people and assets there. To assess seismic hazard and risk in Victoria due to the LRVF-DMF system quantitatively, a fault-source-based probabilistic seismic hazard model was developed in this study and was further extended to a probabilistic seismic loss model for a portfolio of residential wooden houses in Victoria by combining with the building exposure data and seismic fragility functions. The developed seismic hazard model for the LRVF-DMF system considered the synchronous and segmented rupture scenarios of the LRVF-DMF system, as well as the characteristic and truncated exponential magnitude models, allowing the consistent seismic moment release from the fault deformation zone. To investigate the effects of different modelling approaches and their parameters on seismic hazard and risk assessments, a series of sensitivity analyses were performed. Through these assessments, the effects of including the LRVF-DMF system in addition to other seismic sources, such as the Cascadia interface events and deep inslab events, were evaluated to inform decisions related to seismic disaster risk reduction and disaster preparedness in Victoria.

The results from the sensitivity analyses highlight the following conclusions.

- For short-period seismic intensity measures, the seismic hazard contributions from the LRVF-DMF system are not negligible, especially in the low annual probability of exceedance range, and these events should be considered as critical earthquake scenarios for the earthquake impact assessments and disaster preparedness purposes. On the other hand, for long-period seismic intensity measures, the most dominant source is the Cascadia megathrust interface zone, followed by the LRVF-DMF system.
- Overall, among the examined parameters, the influence of the slip rate is the most significant and thus its uncertainty characterization needs to be scrutinized further in conducting a fault-source-based PSHA study. This conclusion is also applicable to the sensitivity analysis results based on the portfolio seismic loss.
- The selections of the segmented versus synchronous rupture scenarios as well as the characteristic versus exponential magnitude models can have pronounced effects on both seismic hazard and risk assessments. The sensitivity analysis results can inform how influential these critical uncertainties are and thus such investigations should be performed.
- The consideration of the LRVF-DMF system results in a 10% to 30% increase in the city-wide building portfolio seismic loss in Victoria. In light of this significant risk potential, although the LRVF-DMF system is not the major contributor of the seismic hazard at the annual probability of exceedance of $4 \times 10^{-4}$ (or 2475-year return period), potential ruptures from this fault source should be considered as one of the critical earthquake scenarios in assessing the adequacy of seismic mitigation and recovery plans that are currently in place for communities in southern Vancouver Island. For such purposes, the integrated use of the advanced seismic hazard and risk models should be promoted at all levels of earthquake risk management in both public and private sectors.

**Author Contributions:** K.G. completed the analyses and writing of the paper. A.S. carried out preliminary analysis of the seismic hazard section as part of his Master's project at Western University. All authors have read and agreed to the published version of the manuscript.

**Funding:** The work was supported by the Canada Research Chair program (950-232015) and the NSERC Discovery Grant (RGPIN-2019-05898).

**Institutional Review Board Statement:** Not applicable.

**Informed Consent Statement:** Not applicable.

**Data Availability Statement:** All relevant data are described and presented fully in the paper.

**Conflicts of Interest:** The authors declare no conflict of interest.

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
