# Peer review of "Fault-Source-Based Probabilistic Seismic Hazard and Risk Analysis for Victoria, British Columbia, Canada: A Case of the Leech River Valley Fault and Devil’s Mountain Fault System"

_sustainability, doi:10.3390/su13031440_

Round 1
Reviewer 1 Report
This paper presents an extensive study of scenario earthquakes in two fault systems and loss estimation. This contribution to the Canadian risk study is highly appreciable. This manuscript extensively covered the issue of seismic risk in the region, considering two faults and scenario earthquakes in the area. In addition, it has also presented the risk of buildings, focusing on timber-framed structures. The approach and methodology are scientifically sound and it is novel for the area of the study. English level is good.
Thank you.
Author Response
Please see the uploaded PDF file.

Reviewer 2 Report
Line 73 and 81. Could it be possible to compare the difference on the same parameter (PGA or/and SA(0.2s))?
Line 138-139. CIS and GTP are not defined.
Line 142. PGT is not defined
Line 149. Why the “>=” before 0.2-0.3?
Line 158. What are the arguments to adopt the fault geometry from this author either than another one?
Line 163. What is the uncertainty of the magnitude estimates?
Line 247. It could be useful for the reader to show in Figure 2 the CIS, GTP and PGT source areas since it is mentioned in the text.
Figure 5. It is not easy to distinguish the Li et al. and NRCan curves. The use of 2 different colors could help.
Figure 15b. the range of built years attributed to the different house value is not coherent with the text explaining this figure.
Figure 16. could you explain the difference between symbols and associated line? how is derived the line from the symbols?
How do you estimate the Vs30 value at each grid point? by averaging the values within each cell?
Author Response
Please see the uploaded PDF file.

Reviewer 3 Report
The goal of the present manuscript is the Fault-source-based Probabilistic Seismic Hazard and Risk Analysis for Victoria, British Columbia, Canada: A Case of the Leech River Valley Fault and Devil’s Mountain Fault System.
Overall, I think this is a really interesting paper with enough data and sufficient and informative figures.
My main concern is the layout of the paper. It does not follow a more typical structure (Introduction-Study Area-Methodology-Results-Discussion-Conclusion), but a different structure. Generally that would not be an issue, but your case though I believe that you should reorganize some sections.
You have so many data to illustrate that eventually, it is not easy to follow. In my opinion a more compatible structure where you introduce the reader to the problem and the study area, mention all the followed methodologies, then all your results from the applied models and then the discussion part, could help the reader to understand the manuscript.
Again, I believe that this is a nice paper with some really interesting results and I do not mean to rewrite it but simple perhaps reorganize some parts.
Within the attached manuscript you will find my comments in detail.

Author Response
Please see the uploaded PDF file.

Reviewer 4 Report
The paper presents a fault-source-based seismic hazard model for the Leech River Valley Fault (LRVF) and the Devil’s Mountain Fault (DMF) in southern Vancouver Island, British Columbia, Canada.
To evaluate the effects of including these faults in probabilistic seismic hazard analysis and city-wide seismic loss estimation for Victoria, a comprehensive sensitivity analysis is conducted by considering different fault rupture patterns and different earthquake magnitude models, as well as variations of their parameters.
However, the paper needs to be improved in a major way. especially the plagiarism of the author with his previous work is more than the allowed quota.
Once the paper is edited in a major way, it can be considered for publication.
The authors should also improve the qualities of the figures and add enough description.
There should be a critical review of the literature presented, mentioning clealry what are the advantages and disadvantes of the previous work.
A significance section should be included, clearly highlighting the significance of this study and comparision with existing studies.
Author Response
Please see the uploaded PDF file.

Round 2
Reviewer 4 Report
The authors have addressed my concerns, Now the paper can be accepted for publication.